



# Effects of $NO_x$ and seed aerosol on highly oxygenated organic molecules (HOM) from cyclohexene ozonolysis

Meri Räty[1], Otso Peräkylä[1], Matthieu Riva[1,2], Lauriane Quéléver[1], Olga Garmash[1], Matti Rissanen[1,3], and Mikael Ehn[1]

[1]Institute for Atmospheric and Earth System Research / Physics, Faculty of Science, University of Helsinki, Finland
[2]Univ Lyon, Université Claude Bernard Lyon 1, CNRS, IRCELYON, F-69626, Villeurbanne, France
[3]Aerosol Physics Laboratory, Physics Unit, Tampere University, Tampere, Finland

**Correspondence:** Meri Räty (meri.raty@helsinki.fi) and Mikael Ehn (mikael.ehn@helsinki.fi)

**Abstract.** Cyclohexene ($C_6H_{10}$) is commonly used as a proxy for biogenic monoterpenes, when studying their oxidation mechanisms and secondary organic aerosol (SOA) formation. The ozonolysis of cyclohexene has been shown to be effective at producing highly oxygenated organic molecules (HOM), a group of molecules known to be important in the formation of SOA. Here, we provide an in depth look at how, on a molecular level, the HOM formation and fate changed with perturbations

from $NO_x$ and seed particles.

HOM were produced in a chamber from cyclohexene ozonolysis, and measured with a chemical ionisation mass spectrometer (CIMS) using nitrate ($NO_3^-$) as reagent ion. As high-resolution CIMS instruments provide mass spectra with numerous ion signals and a wealth of information that can be hard to manage, we employed a primarily statistical approach for the data analysis. To utilise as many individual HOM signals as possible, each compound was assigned a parameter describing the

quality of the observed signal. These parameters were then used as weights or to determine the inclusion of a given signal in further analyses.

Under unperturbed ozonolysis conditions, known HOM peaks were observed in the chamber, including $C_6H_8O_9$ as the largest HOM signal, and $C_{12}H_{20}O_9$ as the largest "dimer" product. With the addition of nitric oxide (NO) into the chamber, the spectrum changed considerably, as expected. Dimer product signals decreased overall, but an increase in dimers with nitrate

functionalities was seen, as a result of $NO_3$ radical oxidation. The response of monomer signals to NO addition varied, and while nitrate-containing monomers increased, non-nitrate signals either increased or decreased, depending on the individual molecules.

The addition of seed aerosol increased the condensation sink, which markedly decreased the signals of all low-volatility compounds. Larger molecules were seen to have a higher affinity for condensation, but a more detailed analysis showed that

the uptake was controlled mainly by the number of oxygen atoms in each molecule. All non-nitrate compounds with at least 7 oxygen atoms were observed to condense onto the seed aerosol at close to equal rates. Nitrates required higher mass and higher oxygen content to condense at similar rates as the non-nitrate HOM. A comparison to experiments with $\alpha$-pinene reported earlier, showed quite a similar relationship between elemental composition and volatility, although products from $\alpha$-pinene ozonolysis appeared to require slightly higher oxygen numbers for the same decrease in volatility. In addition, two models





developed for predicting volatilities of volatile organic compound (VOC) oxidation products were tested on the ozonolysis products of cyclohexene.

## 1   Introduction

Secondary organic aerosol (SOA) plays an important role in the climate system, and its formation and growth has hence been a major focus of research in the recent years (Shrivastava et al., 2017). Through oxidation, some of the initially volatile

organic species can be transformed into low volatility compounds (Jimenez et al., 2009), including highly oxygenated organic molecules (HOM), that can then be readily taken up in aerosol formation (Ehn et al., 2014; Bianchi et al., 2019).

Many of the hydrocarbons emitted to the atmosphere have natural sources, mainly in vegetation (Laothawornkitkul et al., 2009). The oxidation of some of these biogenic volatile organic compounds (BVOCs) have been linked to high SOA yields, for example monoterpenes ($C_{10}H_{16}$) (Tunved et al., 2006; Ehn et al., 2014; Jokinen et al., 2015). Notably, the monoterpenes

that have a carbon ring structure with a double bond (i.e., endocyclic alkenes) can be oxidised by ozone without fragmentation, producing a larger oxygenated molecule (Chuong et al., 2004; Jokinen et al., 2015), often resulting in a higher SOA-forming potential.

The lower the volatility of an oxidation product is, the more it has a tendency to condense onto existing aerosol particles (Kroll and Seinfeld, 2008). In addition to typically having a relatively large molecular mass, low volatility products generally

have high oxygen content. More specifically, low volatility has typically been linked to the oxygen to carbon ratio O:C (Aiken et al., 2008). However, there have also been implications that the relationship between low volatility and O:C ratio might not always be as straightforward as was previously thought (Kurtén et al., 2016). Volatility depends on molecular functionality, and products with more hydrogen-bonding functional groups have been linked to lower volatilities (Donahue et al., 2013). In addition to simple condensation, reactive uptake may also be responsible for some of the growth of SOA particles (Zhang et al.,

45  2015).

The most emitted monoterpene in the atmosphere is $\alpha$-pinene, contributing to SOA formation and growth especially in the Boreal regions (Spanke et al., 2001; Ehn et al., 2014; Jokinen et al., 2015). As most monoterpenes, $\alpha$-pinene has a relatively complex structure. Therefore, the simple symmetrical cyclohexene ($C_6H_{10}$) molecule has regularly been used as its surrogate in SOA studies (e.g., Chuong et al., 2004; Rissanen et al., 2014). As the carbon ring is the common site for oxidation in BVOCs

with endocyclic double bonds (Rissanen et al., 2014), the simple cyclohexene can also provide insight into the oxidation pathways of larger molecules.

In this work, we studied cyclohexene ozonolysis in an environmental chamber, with the main focus of understanding HOM formation and fates under different perturbations. We added NO to the chamber in order to compare the change in HOM formation pathways. In order to test the volatility of the HOM products, we also added seed aerosol to the chamber. We

compare our findings with similar studies performed on $\alpha$-pinene, to test how well cyclohexene can work as a surrogate for monoterpenes with an endocyclic double bond, and how applicable earlier volatility parametrisations are on the cyclohexene system.





## 2 Methods

### 2.1 Chamber conditions and instrumentation

The measurements were conducted during a 5-day-period in the COALA chamber which is described in detail elsewhere (Riva et al., 2019a; Peräkylä et al., 2020). The chamber is 2 $m^3$ in volume and has fluorinated ethylene propylene (FEP) as its wall material. The cyclohexene ozonolysis was performed under dark conditions. Temperature in the chamber was 27 ± 2 °C, and relative humidity 29±2%. Mass flow controllers (MKS, G-Series, 0.05-50 Lpm, Andover, MA, USA) were used to control the injections of all reactants. Ozone was generated from purified air with an ozone generator (Dasibi 1008-PC), and

cyclohexene was produced by flowing air through a glass bubbler with liquid cyclohexene. Both were continuously injected into the chamber, together with purified air (AADCO model 737-14, Ohio,USA). The resulting steady state concentrations of ozone was approximately 18 ppb, while a rough estimate for cyclohexene concentration was about 100 ppb. As the concentration of cyclohexene was not measured it was estimated from the difference in ozone concentration with and without cyclohexene (with rate constant by Stewart et al., 2013). The average chamber residence time was approximately 50 minutes. Ozone and $NO_x$

levels were monitored with gas analysers (UV photometric ozone monitor, Model49p, Thermo Environmental Instruments; NO-$NO_2$-$NO_x$ analyser, Model 42i, Thermo Fisher Scientific). Concentrations of all substances in the different experiments are presented in Table 1.

| Compound | Concentration |
|---|---|
| $O_3$ | 18 ppb [*] |
| $C_6H_{10}$ | 100 ppb |
| NO | 0 ppb[†] |
| | 1 ppb[†] |
| | 3 ppb[†] |
| | 10 ppb[†] |
| 80 nm ABS particles | 0 $\mu$gm$^{-3}$ |
| | 9 $\mu$gm$^{-3}$ |

**Table 1.** The reactant, oxidant and particle concentrations during the conducted experiments ([*]steady state concentration when no NO input, [†]input concentrations of NO which mostly converted quickly into $NO_2$)

$NO_x$ experiments were conducted by injecting NO into the chamber at concentrations shown in table 1. As NO quickly reacts with ozone to form $NO_2$, the resulting NO/$NO_x$ ratio in the chamber was fairly low, approximately 3±1% during

highest injection rates. For most of the time, the concentration of NO in the chamber was not high enough to exceed the detection limit of the used analyser. Therefore, only the total $NO_x$ was used in the analyses.

The condensation of vapours was studied with a 9 $\mu$g m$^{-3}$ loading of ammonium bisulfate (ABS) (($NH_4$)$HSO_4$) particles that were produced by nebulising aqueous ammonium sulfate solution with sulfuric acid. They were dried and size selected



(80 nm) with a differential mobility analyser (DMA) before injection. The condensation sink was calculated for sulfuric acid,
from aerosol size distribution data (Maso et al., 2005) measured with a Differential Mobility Particle Sizer (DMPS, Aalto et al.,
2001). Experiments with ammonium sulfate were also carried out, but in these the loading was not sufficiently high to create
a significant change that would allow for conclusions to be drawn. Therefore, these were left out of the analysis. Due to the
acidity of the ammonium bisulfate aerosol, it should be noted that there is a possibility that a fraction of the overall sink might
be a result of reactive uptake (Zhang et al., 2015). However, in their similar experiments with $\alpha$-pinene, Peräkylä et al. (2020)
did not observe the uptake of HOM to differ significantly between AS and ABS seeds.

Oxygenated gas phase products were monitored by a Chemical Ionisation Atmospheric Pressure interface Time-of-flight
mass spectrometer (CI-APi-TOF, TOFWERK AG, Aerodyne, Junninen et al., 2010; Jokinen et al., 2012). It consists of three
components, a chemical ionisation inlet (CI), an atmospheric pressure interface (APi) and the Time-Of-Flight (ToF) mass
spectrometer. Nitrate ions produced by subjecting an air stream containing nitric acid ($HNO_3$) to soft x-rays, are used to charge
the sample molecules drawn from the chamber into the CI inlet. After charging, ions are directed into the APi. Quadruple ion
guides and an ion lens stack focus and guide the ions through the APi chambers where the air is pumped out, with the pressures
ultimately reaching $10^{-6}$ mbar in the TOF, where the ions' mass-to-charge ratio is determined. Nitrate ions have been shown
to cluster readily with HOM molecules (Ehn et al., 2014; Bianchi et al., 2019). The clustering is not necessarily equally
efficient between all compounds however, and signals are therefore not necessarily directly representative of the actual relative
concentrations of the compounds. This selectivity of the nitrate ions has been described by e.g. Hyttinen et al. (2015).

The CI-APi-TOF mass spectrometry data was analysed with a Matlab (R2016a) based toolbox, tofTools (R607) (Junninen
et al., 2010). In further analysis the signals at different times were made comparable by normalising all time series by di-
viding them with the sum of the three reagent ion time series ($NO_3^-$, $HNO_3NO_3^-$, $(HNO_3)_2NO_3^-$, Jokinen et al., 2012). As
quantification of HOM remains challenging (Riva et al., 2019b), we opted for analysing only the relative changes to HOM
concentrations in this work.

## 2.2 Peak Fit Analysis

tofTools fits peaks based on a given list of compounds. Some of the compounds could potentially have very similar masses,
and fitting them correctly can sometimes prove challenging for an automatic routine (Cubison and Jimenez, 2015; Zhang et al.,
2019). Problems can also arise if the spectra are noisy, or if an observed ion is not listed in the compound list but something
with a close mass is. These types of issues may result in fitting a peak either too small or too large.

Including potentially false signals would be undesirable in the analysis. Individually estimating the quality of each peak fit
at each time point, is extremely time-consuming and presents an inherent challenge for high-resolution mass spectral analysis.
This motivated us to attempt to use a quantitative measure to describe the quality of the automatically generated fits. By
assigning this variable for each fit, signals with poorly fitted peaks could be easily identified, and down-weighted or removed
from the analysis.





The variable, which we will call the FitFactor, was used to quantify how well the fitted peak matched the spectra. The residual area, i.e. the difference between the spectral peak and the sum of all fitted peaks, is compared to the surface area of the fitted peak of interest. This residual-to-peak ratio is subtracted from unity, and the reached value is the FitFactor of the peak fit.

$$FitFactor = 1 - \frac{A_{residual}}{A_{peak}}$$

Thus, the better a peak fit is, the closer the value is to 1 (Figure 1 a.). Values close to zero or even negative values are conversely associated with poor fits (Figure 1 d.). Something in between can be reasonable (1 c.), but as the value decreases, uncertainty generally increases. Each compound has a single FitFactor value assigned to it. Spectra with signal below 50% of the time series maximum were ignored, because analysing for example a fit of a nitrate compound is not relevant during times when no NO is being added to the chamber. The final compound FitFactor was the median of individual FitFactor calculations from the

rest of the spectra.

By applying the FitFactor, a majority of the very poorly fitted peaks, such as the one shown in Figure 1 d., can easily be identified and removed from the analysis. However, some compounds may still have a relatively good FitFactor, despite the fit being somewhat questionable, like in the case of $C_5H_9NO_5NO_3^-$ shown in Figure 1 b. These types of fits are especially tricky. The automatic peak fitting routine is capable of producing a sum peak that matches the spectra relatively well, which translates

into a high FitFactor value. However, as so many peaks (3 compound peaks, 1 large isotope peak) have been closely fitted onto the same spectral peak, there is a great deal of uncertainty related to the individual peak fits. Thus, the FitFactor might not be optimal for this type of multiple fit problems, but thanks to the selectivity of the nitrate CI-APi-TOF, such peaks are not very prevalent in the spectra. The problem could be avoided by setting a minimum separation between fitted peaks, which we did not do in this work. Since the peak list was generated semi-automatically by simply listing all compositions within chosen

element number limits, setting a minimum FitFactor value limit, is however, a helpful way to eliminate most, although not all, of the poorest fits. The analysis can then be focused on the most reliable signals.

### 2.3   Peroxy radical chemistry and product formation

The oxidation and HOM formation pathways of cyclohexene ozonolysis is described in detail by Rissanen et al. (2014). In this section, we briefly outline the main steps considered when interpreting the results in this work. First, ozone oxidises

cyclohexene by attaching itself on the double bond, leading to ring-opening. After rearrangement, loss of an OH radical, and an addition of an $O_2$ molecule, a primary peroxy radical ($RO_2$) of the chemical composition $C_6H_9O_4$, is formed (Berndt et al., 2015). Peroxy radicals can undergo several steps of autoxidation (Crounse et al., 2013), producing an increasingly more oxygenated $RO_2$ radical, that may eventually terminate into a highly-oxidized end-product (i.e., HOM). However, the ultimate fate of the primary and potential subsequent $RO_2$ radicals will depend on the exact conditions under which the oxidation takes

place. This will, subsequently, greatly impact the final product molecules and their properties, such as volatility. Therefore, we outline the relevant main pathways below.







**Figure 1.** Examples of determining the FitFactor from the residual-to-peak ratio of four different compounds. The title compounds are indicated by green lines and shading. The FitFactor value for each is shown in the upper right corner of the subplots. a) A good fit results in a FitFactor close to 1. b.) Accurate determination of the fit quality is tricky, when this many compounds and/or isotopes are fitted to the same spectral peak. Automatic fitting can artificially produce a good sum fit, which translates into a good FitFactor, but single compound fits may lack credibility. c.) The two fitted peaks match the spectra relatively well, resulting in a reasonably good FitFactor for the bigger peak. d.) A very poor fit results in low or even negative FitFactor.

Before reaching termination, some radicals may be susceptible to a decrease in carbon number following a loss of a CO group (Mereau et al., 2001; Rissanen et al., 2014):

$$C_xH_yO_z \rightarrow C_{x-1}H_yO_{z-1} + CO \tag{R1}$$





Unimolecular termination of $RO_2$ typically takes place through the loss of a hydroxyl radical (OH) (Rissanen et al., 2014):

$$C_xH_yO_z \rightarrow C_xH_{y-1}O_{z-1} + OH \tag{R2}$$

A loss of $HO_2$ is another previously suggested mechanism that has since been shown not to be likely under atmospheric temperatures (Hyttinen et al., 2016). Bimolecular reactions between $RO_2$ and $HO_2$, or between two $RO_2$ radicals also produce closed-shell species (Orlando and Tyndall, 2012; Rissanen et al., 2014). The former can typically produce ROOH molecules 150 (Eq. (R3)), whereas the latter can lead to a production of a dimer molecule (ROOR) (Eq. (R4)) or two monomers (ROH and $R^\diamond CHO$) (Eq. (R5)). Alternatively, bimolecular reactions can also produce reactive open-shell alkoxy radicals (RO) (Eq. R6).

$$RO_2 + HO_2 \rightarrow ROOH + O_2 \tag{R3}$$
$$RO_2 + R'O_2 \rightarrow ROOR' + O_2 \tag{R4}$$
$$RO_2 + R'O_2 \rightarrow ROH + R^\diamond CHO + O_2 \tag{R5}$$
$$RO_2 + R'O_2 \rightarrow RO + R'O + O_2 \tag{R6}$$

Peroxy radicals are also a part of the chemistry of pollutants such as $NO_x$ (i.e. NO and $NO_2$). The reactions between $NO_x$ and $RO_2$ usually take one of the following forms (Orlando and Tyndall, 2012):

$$RO_2 + NO \rightarrow RO + NO_2 \tag{R7}$$
$$RO_2 + NO \rightarrow RONO_2 \tag{R8}$$
$$RO_2 + NO_2 \rightleftharpoons RO_2NO_2 \tag{R9}$$

Typically only reaction (R8) is significant in producing closed-shell end-products, as products formed in reaction (R9) are thermally unstable, unless under very cold conditions (Orlando and Tyndall, 2012). Under short reaction times a high concentration of $NO_2$ has been shown to lead to dimer suppression and formation of highly oxidised nitrates in the cyclohexene 165 ozonolysis system (Rissanen, 2018). However, here the longer reaction times and the much lower $NO_x$ concentration are not likely to allow for a significant nitrate pool from $NO_2$ reactions due to the equilibrium of reaction R9 residing strongly on the reactant side under our experimental conditions. A fraction of $NO_2$ radicals may react with ozone to produce $NO_3$, which is also one of the main oxidising species in the atmosphere. This oxidation of a BVOC compounds by $NO_3$ radicals produces nitrogen containing peroxy radicals, which can form both monomer and dimer species (Yan et al., 2016).

## 3 Results and Discussion

Figure 2 shows a HOM spectrum from cyclohexene ozonolysis (without NO addition), with ten of the biggest observed oxidation product peaks labelled. Other large peaks shown are the nitrate dimer at 188 Th, and an unidentified peak at 234 Th. Strong signals of the HOM monomers $C_6H_8O_7$ and $C_6H_8O_9$ are observed, which has been typically reported also in other



**Figure 2.** An example 10-minute average spectrum of cyclohexene ozonolysis. Ten of the biggest product peaks (darker green) are labelled with their chemical composition. All products are detected as cluster with $NO_3^-$, the mass of which is included in the masses shown in the x-axis.

studies (Rissanen et al., 2014; Berndt et al., 2015). A number of the largest signal peaks are from relatively little oxidised

compounds, with 4-5 oxygen atoms. Based on their elemental formulas, these species are most likely semi volatile organic compounds (SVOC) (Peräkylä et al., 2020), which might accumulate in the chamber making them much more abundant than the low-volatile HOM, which are efficiently removed by condensation. This might explain the large signals even though the charging efficiency of such compounds is generally low (Hyttinen et al., 2015). In addition to the monomers, also one dimer signal, $C_{12}H_{20}O_9$, reaches the top ten. One potential formation pathway for this dimer is a bimolecular reaction (R4) between

radicals $C_6H_9O_8$ and $C_6H_{11}O_3$. The latter is the primary peroxy radical formed form OH-oxidation of cyclohexene (Berndt et al., 2015).






**Figure 3.** An example 10-minute average spectrum of cyclohexene ozonolysis in the presence of $NO_x$ (approx. 9 ppb). Ten of the biggest product peaks (darker blue) are labelled with their chemical composition. All products are observed as clusters with $NO_3^-$, the mass of which is included in the masses shown in the x-axis.

### 3.1 Influence of NO on cyclohexene oxidation products

Injecting NO into the chamber has a significant impact on the chemistry within, which consequently leads to changed product signals. Although only NO mainly has an impact on the radical chemistry (see Eq. (R7)-(R9)), due to the aforementioned technical limitations leading to a lack of separate measurements of NO, we compare signals to the measured $NO_x$. An example spectrum during a $NO_x$ experiment is shown in Figure 3. The ten biggest signals are otherwise the same as in the absence of $NO_x$ (Figure 2), except for the dimer signal ($C_{12}H_{20}O_9$) having dropped to a fraction of the original and thus off the list, while a nitrate signal, $C_6H_9NO_9$ (Eq. R8), has replaced it. The monomer signal peaks are also on average larger than in Figure 2.





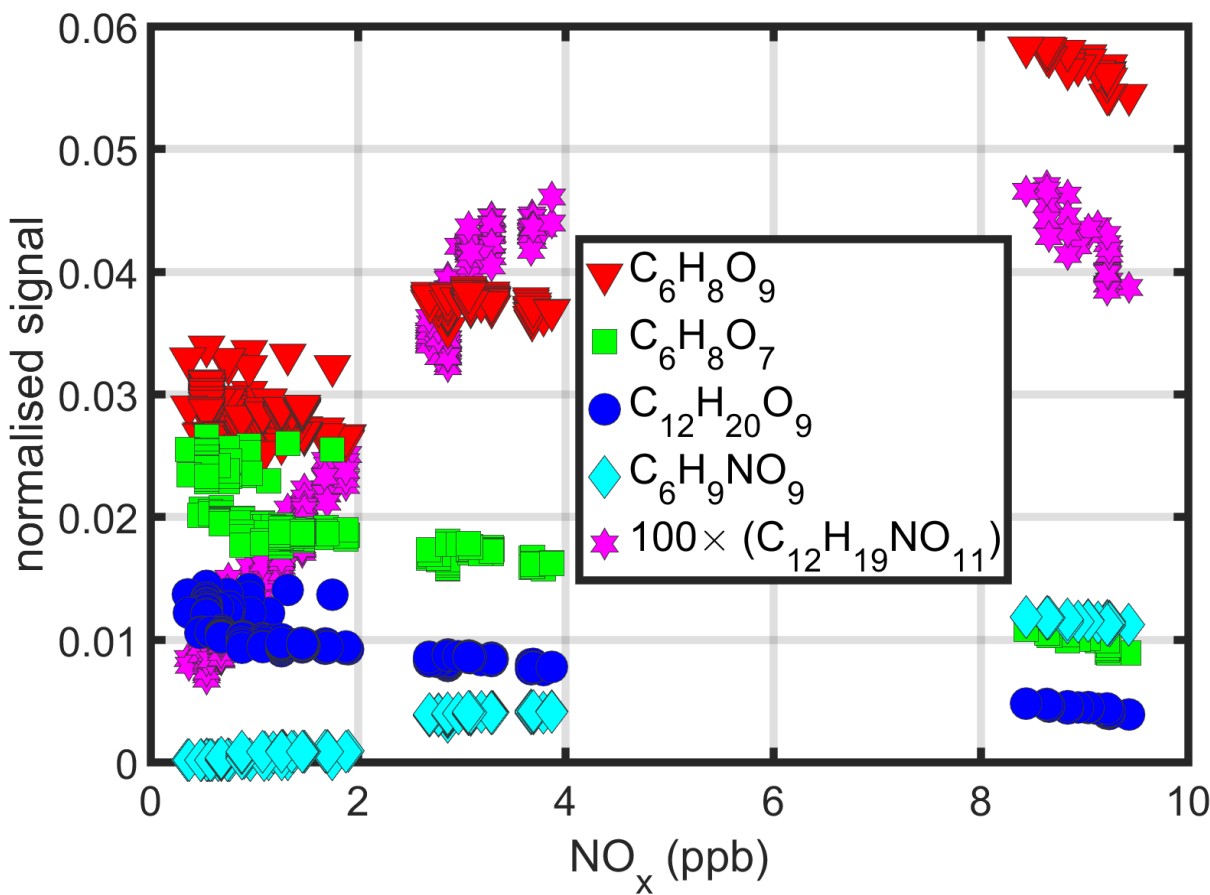

**Figure 4.** Examples of signal changes due to increased $NO_x$ levels in the chamber. Shown are examples of two differently behaving monomers and a dimer without nitrogen, as well as a nitrogen-containing monomer and dimer. The last one had a signal so small that in the Figure it is multiplied by a 100. No measurements were done with $NO_x$ concentrations between 4 and 8 ppb

A few examples of changes in product signals with increasing $NO_x$ are shown in Figure 4. Species with comparable signal sizes are purposely selected (except for the nitrogen-containing dimer that has a much smaller signal). Shown are the increasing monomer $C_6H_8O_9$ signal, decreasing $C_6H_8O_7$ monomer signal, decreasing $C_{12}H_{20}O_9$ dimer signal and increasing $C_6H_9NO_9$ nitrate and $C_{12}H_{19}NO_{11}$ nitrogen-containing dimer signals. When $NO_x$ is present in the chamber, the occurrence of $RO_2 + NO$ reactions (Eq.(R7)-(R8)) reduce the likelihood for other bimolecular reactions (Eq. (R3)-(R6)).

Nitrogen containing species are only expected to form during the $NO_x$ experiments, as they are either formed in bimolecular reactions between $RO_2$ and $NO_x$ (Eq. Eq. R8-R9), or from radicals oxidised by $NO_3$. Nitrogen containing dimers form in bimolecular reactions (Eq. (R4)), where one of the reacting radicals has been oxidised by a nitrate radical.

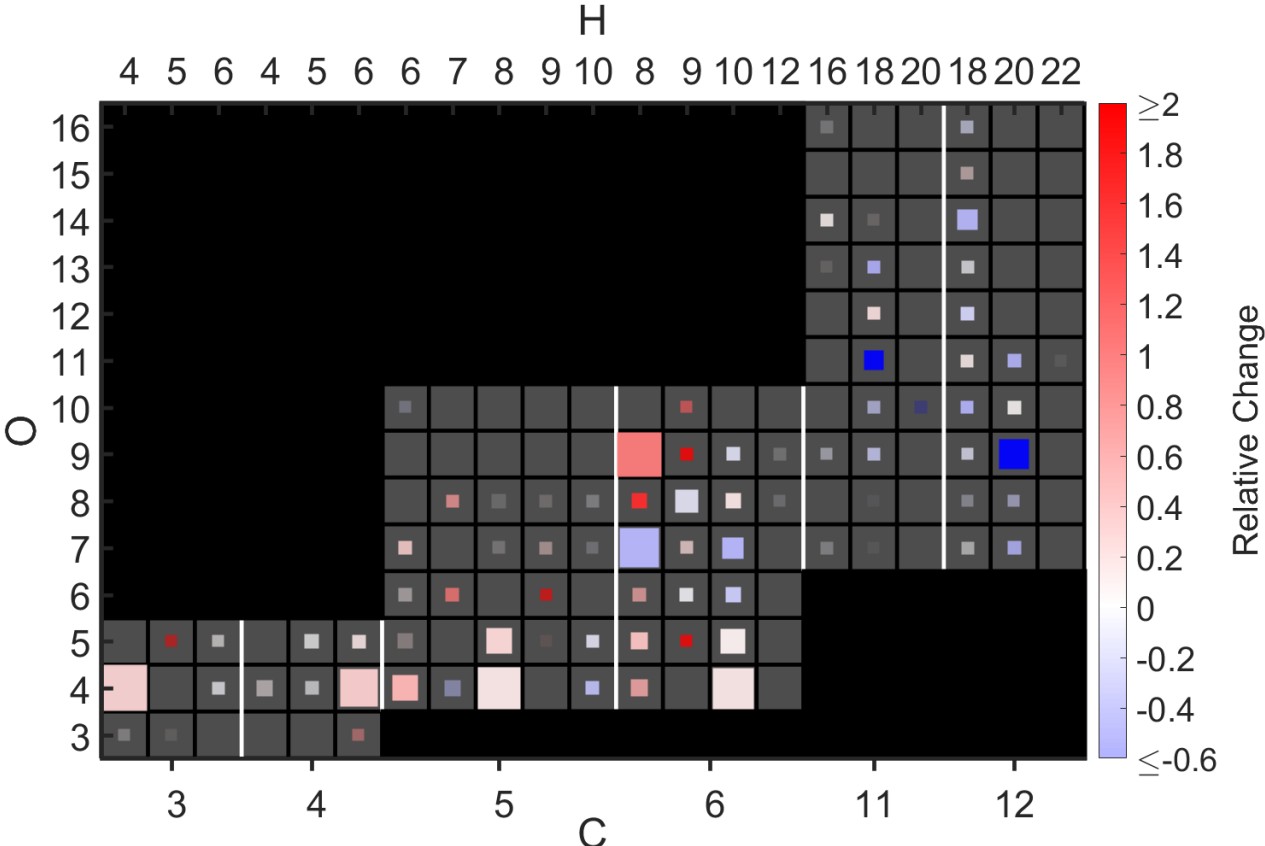

**Figure 5.** The effect of $NO_x$ on observed signals of molecules that do not contain nitrogen. Signals from when $NO_x$ concentration was 9ppb ("high $NO_x$"), are compared to signals during no NO injection ("zero $NO_x$". Although, small $NO_x$ residual $\leq$1ppb). The bottom x-axis shows the carbon number of each compound, with the white vertical lines separating the different C-atom numbers. The top x-axis shows the hydrogen number, and the y-axis the oxygen number. Compounds with an odd number of H-atoms are radicals. Marker colour corresponds to the relative change in the signal size, when moving from zero $NO_x$ to high $NO_x$, marker size describes the size of the signal at zero $NO_x$ and the marker transparency describes the quality of the peak fit (if FitFactor=1, marker is fully opaque; if FitFactor$\leq$0, marker is fully transparent). A few products had values outside the colour axis limits. Columns with no appreciable signals were omitted from the figure.

For a broader overview of signal changes in response to $NO_x$ loading, Figure 5 shows the relative change of all non-nitrate products signals (nitrate-containing molecules are presented in Fig. 6 and discussed later) that were fitted. Signals with 9 ppb of $NO_x$ were compared to signals without $NO_x$ addition. All compounds with even hydrogen numbers are closed shell products, whereas ones with an odd number are radicals.

The results in Figure 5 paint a complicated picture on the formation of the non-nitrate compounds. While some features are expected and easy to explain, others remain less clear. Here we summarise some aspects of the findings. As adding NO into the chamber leads to peroxy radicals reacting with NO (Eq. (R7)-(R8)), other bi-molecular reactions (Eq. (R3)-R6) consequently





decrease. The most clear effect is the general decrease of almost all dimers, as the $RO_2$ cross reactions forming them decrease. For monomers, the effects are more complicated, with some increasing and some decreasing. For example, the signals of the two largest HOM, $C_6H_8O_7$ and $C_6H_8O_9$ showed opposite behaviour, with the former decreasing and the latter increasing. Speculating the reason for this difference is difficult, especially as the reaction rate constants and different branching ratios of reactions between different combinations of $RO_2$ are unknown, and can vary by orders of magnitude depending on the specific structures (Shallcross et al., 2005; Berndt et al., 2018). Most of the $C_6H_{10}O_x$ compounds showed a decrease, which may be explained by a potential decrease in the bimolecular reactions R3 and (R5). The $NO_x$ will also function as a sink for OH, although the cyclohexene should still dominate the OH reactivity in the chamber even at the highest $NO_x$ concentration. Most of the less oxidised SVOC showed relatively small changes.

The increase of the observed nitrate signals under high $NO_x$ (9 ppb) conditions is shown in Figure 6. Many compounds were associated with an unexpectedly high signal even outside of $NO_x$ experiments, and the size of this background signal is also shown in the figure. We opted to not simply subtract the background, i.e. the signal at zero added NO, because in some cases the background was much larger than the signal increase, which would have caused large uncertainties when subtracting two relatively large numbers. In this way, the size of the background is also visible, and one can identify more clearly which increases are most relevant, i.e. the markers with the thickest yellow lines.

Unsurprisingly, nitrates increased with higher $NO_x$. $C_6H_9NO_9$ was clearly produced the most, and likely formed in the reaction between NO and the radical $C_6H_9O_8$, which is one of the most abundant highly oxygenated $RO_2$ radicals in this system (Berndt et al., 2015) (Eq. (R8)). Nitrogen-containing dimers are also detected. This implies that some of the cyclohexene had been oxidised by $NO_3$, which had led to the formation of nitrogen containing radicals (Yan et al., 2016). These radicals were then terminated by $RO_2$ radicals from ozonolysis reactions, as the observed nitrate-containing dimers only had one nitrate functionality. The most abundant dimer was $C_{12}H_{19}NO_{11}$. It can form in the reaction of the $C_6H_9O_8$ radical, from ozonolysis, together with the primary $RO_2$ radical $C_6H_{10}NO_5$, from $NO_3$ oxidation of cyclohexene.

### 3.2 Condensation properties of cyclohexene oxidation products

The condensation sink of ABS seed aerosol was used to assess species volatilities. Compounds with low volatilities are expected to rapidly condense onto the particle surfaces, whereas the more volatile compounds should remain in the gas phase. The loss rate of the latter is mainly governed by flush out from the chamber, while the low-volatile compounds condense either onto the walls or onto particles ($k_{loss} \approx k_{CS} + k_{wall}$). For a more detailed discussion on the dynamics of the chamber and this method, see Peräkylä et al. (2020). Figure 7 shows the signal response of three species against the condensation sink (CS) of the injected ABS particles. The signals are normalised so that the median of the initial signal value before the seed experiment equals 1. As a result, the fraction that remains following the seed addition is easy to observe. The three compounds in Figure 7 vary only by their oxygen number. The most highly oxygenated vapour, that is, $C_6H_{10}O_8$, condenses most readily and thus can be described as being the least volatile.

The fraction remaining (FR) at a condensation sink of approximately $0.85 \times 10^{-2} s^{-1}$ was calculated for all compounds fitted in the mass spectra. These are plotted in Figure 8 against the mass of each compound. The methodology here closely follows

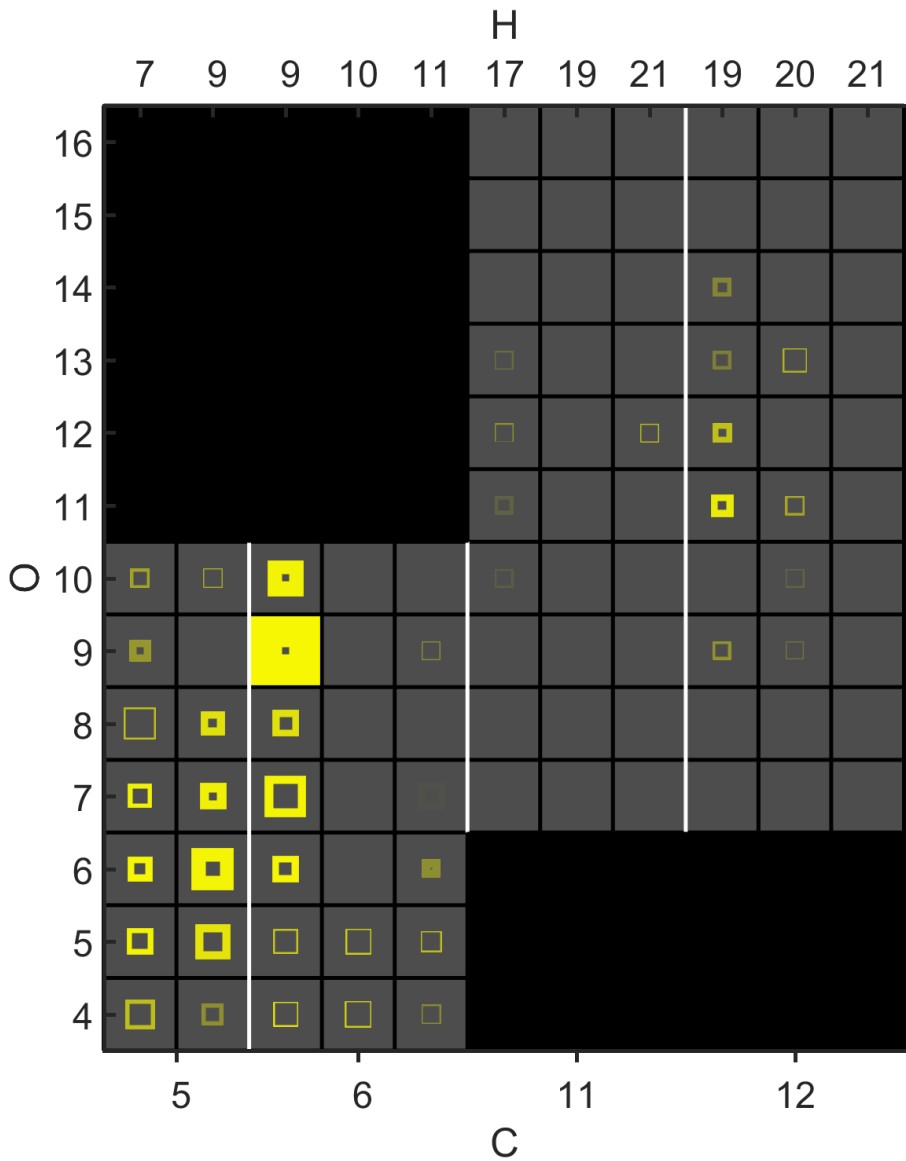

**Figure 6.** Signal changes in N-containing molecules upon $NO_x$ addition during cyclohexene ozonolysis. The axes and transparency follow the same logic as in figure 5. Yellow squares show the size of the signal under high $NO_x$ (9 ppb) conditions. The "holes" in the square correspond to the size of the background signal, i.e. the signal before $NO_x$ addition. The large background for some peaks may be due to small amounts of $NO_x$ being introduced to the chamber, or that these molecules were residues from chamber wall following earlier experiments.

that outlined by Peräkylä et al. (2020). Only the clearest signals are shown, with the aim to bring only the most meaningful signal into focus while minimising noise. In practice, this means compounds with FitFactor values of at least 0.7, which was





**Figure 7.** Condensation behaviour of three $C_6H_{10}O_x$ species. Signals are normalised so that the median of the signal before the ABS seed experiment is set to 1. Thus, shown on the y-axis is the remaining fraction of the signal of each molecules at a given condensation sink. The dashed line boxes encircle data points between the condensation sink values $0.825 \times 10^{-2} s^{-1}$ and $0.875 \times 10^{-2} s^{-1}$, and the black edged circles show the median of these boxed-in data points.

240    deemed to strike a good balance between quality (see Figure 1) and quantity. Non-nitrate and nitrate products are both plotted, while radicals are not, since condensation is only a minor sink for these species because of their high reactivities with other radicals (Peräkylä et al., 2020).

     In addition to observations, also predictions based on two different models are shown in Figure 8. For both models, the modelled values were scaled to the range of observed remaining fractions, between 0.43 and 1. Estimates in the upper panel

245    are modelled with the parametrisation by Peräkylä et al. (2020),



$$\log_{10}(C^*) = 0.18 \times n_C - 0.14 \times n_H - 0.38 \times n_O + 0.8 \times n_N + 3.1$$

that was originally based on measurements with $\alpha$-pinene oxidation products. $n_C$, $n_H$, $n_O$, and $n_N$, correspond to the numbers of carbon, hydrogen, oxygen and nitrogen atoms in the molecule (charging ion excluded).

The model for volatility used in the lower panel figure is given by Bianchi et al. (2019), although the model we used was further modified to account for nitrate functionalities. In nitrates, 3 oxygen atoms are bonded in the nitrate group, which as a whole reduces volatility by a factor of 2.5. We assumed an activity coefficient ($\gamma$) of 1, so that C*=$\gamma$C$^0$=C$^0$ (Donahue et al., 2011). Thereby, the equation used in this work is

$$\log_{10}(C^*) = (n_{C0} - n_C)b_C - (n_O - 3n_N)b_O - 2\frac{(n_O - 3n_N)n_C}{n_C + (n_O - 3n_N)}b_{CO} - b_N n_N$$

Here the additional constants are $n_{C0} = 25$, $b_C = 0.475$, $b_O = 0.2$, $b_{CO} = 0.9$ and $b_N = 2.5$ (Bianchi et al., 2019). To be able to compare the modelled volatilities to our observations of FR, we converted the C*-values to remaining fractions they would theoretically correspond to under our experimental conditions. The conversion was done similarly to Peräkylä et al. (2020), who fitted a logistic curve $y = \frac{y_{max} - y_{min}}{1 + e^{-3.0(x - 0.13)}} + y_{min}$, where $y_{min}$ and $y_{max}$ are limits for scaling, between FR of model compounds and the common logarithm of their saturation concentration from the ADCHAM model (Roldin et al., 2014). The original conversion equation had to be modified, as the SOA concentration in our experiment was an order of magnitude lower than in the Peräkylä et al. (2020) experiment, which would lead to lesser condensation and higher FR at similar C*. To account for the difference, we shifted the original fit between FR and $\log_{10}(C^*)$ left by 1 (i.e. plugging in $x = \log_{10}(C^*) + 1$), thus adjusting the relationship between FR and C* by a factor of 10. While certainly introducing additional uncertainty, we expect this approach to allow an adequate comparison of our findings with those of Peräkylä et al. (2020) and Bianchi et al. (2019).

Many monomers had remaining fractions above 1, indicating that the signal increased. The likely explanation for this is that these compounds are semivolatile and were still equilibrating with the chamber walls, and thus not in steady-state. Another possibility is that these molecules have condensed phase sources, causing an increased formation during the seed addition. Due to these uncertainties, we do not try to interpret these points further, but note that although some of the molecules with FR<1 may also have seen similar effects, the effects are probably smaller as these molecules are less likely to evaporate from particles or walls.

The higher the mass of the molecule, the more inclined it was to be taken up by the injected particles. The condensation sink caused most of the non-nitrate signals with masses above 250 Th to drop to values less than 50% of the original. Such a high net condensation is a clear indication of a low volatility. At masses below 225 Th, compounds were fairly volatile, as all signals remained either the same or increased. The exception to this are the two small green points around 200 Th, with FR<0.7. These are $C_5H_{10}O_4$ and $C_5H_{10}O_5$, which are less oxidised than molecules typically observed with the nitrate adduct CI-APi-TOF. This may mean that they contain multiple hydroxyl and/or carboxylic acids groups (which may improve their clustering ability with $NO_3^-$) that decrease their volatility more than if the O-atoms were incorporated as carbonyl and hydroperoxide functionalities, which is more typical for HOM.



**Figure 8.** The observed (opaque) and modelled (transparent) decrease in signal during an ABS seed experiment, for non-nitrate (green) and nitrate (blue) products. The top panel compares our observations with the model presented by Peräkylä et al. (2020), while the bottom panel compares to the model by Bianchi et al. (2019). Only compounds with the best peak fits (FitFactor$\geq$0.7) are shown. The circles sizes are based on the relative sizes of the initial signals before the ABS aerosol injection, with the exception of the two biggest signals, which were slightly scaled down. The condensation sink was approximately $0.85 \times 10^{-2} s^{-1}$, meaning that these points correspond to the black edged circles in Figure 7. All molecules were charged with a nitrate ion, which has a mass-to-charge ratio of 62 Th.

Similarly to non-nitrates, the volatility of nitrates decreases with mass. However, the transition from high to low volatility appears to happen at a larger mass than for non-nitrates, which is in line with previously determined group additivity trends of volatility (Pankow and Asher, 2008; Kroll and Seinfeld, 2008). A clear outlier is $C_5H_9NO_5$ (225.04 Th), as the signal





comparison suggests an unexpectedly low volatility for the compound. This might however be an artefact from issues with the

fitting of multiple peaks, as shown specifically for this ion in Figure 1 b.

Both the Peräkylä et al. (2020) and Bianchi et al. (2019) models successfully predict the high volatility of the least oxidised monomers. The predicted volatilities of larger monomers are however in both cases higher than the observed. Both models predict the strongest volatility transition in the region between our observed monomers and dimers, although we observe many of the monomers to condense nearly irreversibly. Peräkylä et al. (2020) model predicts a more gradual decrease in volatility as

a function of mass than the Bianchi et al. (2019) model.

Next, the link between elemental composition and volatility is analysed further. In Figure 9, the fraction remaining is plotted separately against carbon, hydrogen and oxygen numbers, as well as the "effective O:C ratio" $(n_O - 2n_N)/n_C$. The incentive for choosing this ratio, rather than simply O:C, is the same as for the modification of the Bianchi et al. (2019) model, i.e. to account for nitrate functionalities. For non-nitrates this expression reduces to an O:C ratio. For comparison, also $\alpha$-pinene

products from an experiment by Peräkylä et al. (2020) are plotted on the background. The data is from an experiment with dry ABS seed aerosol and $NO_x$. As the reactant concentrations and chamber conditions differed between our and their experiments, the signals are not comparable in magnitude. In Figure 9, the observations have been scaled to the same marker size and FR range. Due to the higher SOA concentration in the Peräkylä et al. (2020) experiment, the remaining fractions of $\alpha$-pinene products are somewhat lower than they would be at our experiment's SOA concentration.

Molecules with the lowest carbon and hydrogen content hardly condensed, while the opposite was true for the molecules with the highest C and H content, mostly associated with the dimers (Fig. 9a and 9b). However, in the regions between these extremes, e.g. around $C_6$ and $H_8$, respectively, molecules range from volatile to effectively non-volatile, and thus purely C- or H-atom content is not a good predictor for volatility. As a function of O-atom content (Fig. 9c), the volatility seems to be much more monotonically decreasing. The transition from non-condensing to condensing occurs at O-atom contents of 6-7 for

cyclohexene HOM, with nitrates again being slightly more volatile than non-nitrates. Finally, figure 9d demonstrates that the effective O:C ratio is the worst predictor for volatility out of the parameters plotted. Despite O:C ratios of organic aerosol often being used as a reference for volatility, for a single molecule it does not have a good correlation. This is to be expected, as high molecular weight dimers require only a relatively small number of oxygen atoms (thus having low O:C) to condense, while small molecules with 1-3 C-atoms are unlikely to condense regardless of their O:C ratios. In all figures (9a-d), the $\alpha$-pinene

and cyclohexene products showed similar behaviour.

## 4 Conclusions

We have investigated the formation and fates of cyclohexene oxidation products, with a focus on the most oxidised species. These highly oxygenated organic molecules (HOM) were measured with a CI-APi-TOF mass spectrometer. A statistical approach was utilised for evaluating the quality of the identification of different ions in the mass spectra. In this approach, a

so-called FitFactor parameter was set up to compare the spectral residual and the peak area. By this simple method, evaluating





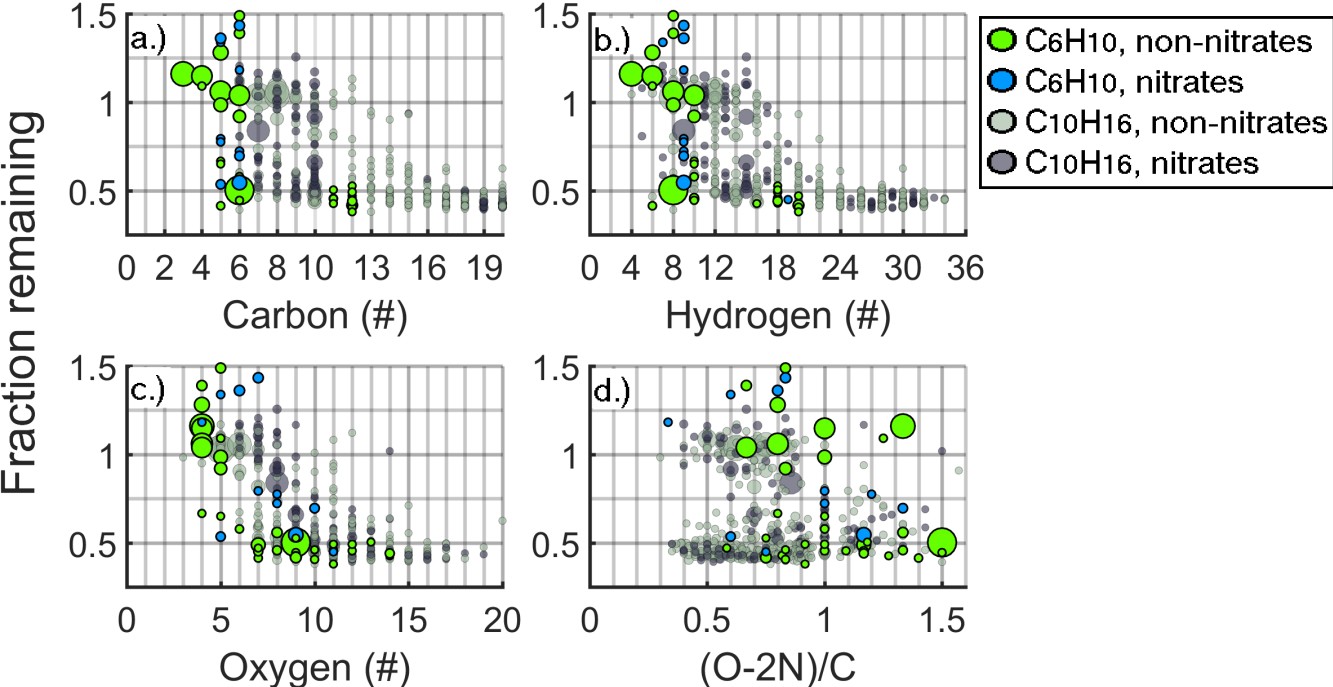

**Figure 9.** Fraction remaining (FR) after seed addition, as a function of elemental composition. This figure is similar to Figure 8, except that here the FR is plotted against the a.) carbon, b.) hydrogen and c.) oxygen content of the compounds, and against d.) the effective O:C ratio (O-2N)/C. In addition to the cyclohexene ($C_6H_{10}$) oxidation products, $\alpha$-pinene ($C_{10}H_{16}$) ozonolysis products from a seed experiment by Peräkylä et al. (2020) are also included. As the signals between the two studies are not directly comparable, the $\alpha$-pinene products have been scaled to the same limits of FR, and to the same range of marker sizes.

thousands of peak fits was fast and feasible. Ultimately, this allowed discarding or down-weighing signals with poorly fitted peaks.

Perturbations to product signals by $NO_x$ were analysed species-by-species, revealing that $NO_x$ enhanced the formation of most of the observed monomer ($C_3$-$C_6$) species, although some also experienced an opposite effect. These effects are likely driven by increased termination of $RO_2$ radicals by NO, leading to more alkoxy radicals, but with the current data, the detailed reasons for the individual differences can only be speculated. However, more easily interpretable effects were also observed, namely the formation of organic nitrates, and the decrease of dimer ($C_{11}$-$C_{12}$) species, due to enhanced $RO_2 + NO$ reactions. This reaction, taking place at the expense of $RO_2$ cross reactions, caused a decrease in dimers without a nitrate functionality. Dimers containing a nitrate group conversely experienced a small increase. This is likely due to $NO_3$ radical (formed from $NO_2 + O_3$) oxidation forming $RO_2$ with nitrate functionalities, which can the react with other $RO_2$ to form dimers as long as the NO concentration is not high enough to completely dominate the $RO_2$ termination in the system. Monomer nitrates, on the other hand, see a monotonically increasing trend as more NO is added to the chamber, as expected.



The volatility of the oxidation products was probed in experiments where ammonium bisulfate seed aerosol was added to the chamber. This resulted in a clear decrease in most HOM signals, with all non-nitrate products heavier than 250 Th condensing

to the particles at nearly equal rates. Nitrates were slightly more volatile than non-nitrates with similar masses. Besides mass, a break-down of the elemental composition of the different products indicated that the most significant factor determining the volatility of the observed cyclohexene oxidation products is the oxygen number. Products having an oxygen atom content of 7 or more, were all at the lowest end of the observed volatility range. Carbon and hydrogen atom content also correlated with volatility, but were clearly more ambiguous indicators for volatility than the oxygen content. For example, molecules with 5

or 6 C-atoms can range anywhere from volatile to effectively non-volatile. We also found that the O:C ratio is by itself not a good measure for volatility, as molecules like $C_3H_4O_4$ have a much higher volatility than e.g. $C_{12}H_{20}O_8$, despite having an O:C ratio twice as high.

There have been a few models presented recently for predicting the volatility of a compound based on its elemental composition. We tested our observed volatilities against two, the model by Bianchi et al. (2019) with a modification for nitrates,

and another by Peräkylä et al. (2020). They are based on larger initial reactants than cyclohexene, such as $\alpha$-pinene. While both could predict the generally higher volatility of monomers compared to dimers, they were not able to produce the observed irreversible condensation of larger monomers. This highlights the difficulty in estimating volatilities with simple parametrisations. As cyclohexene is used as a surrogate for monoterpenes like $\alpha$-pinene, we also compared the volatilities of HOM from $\alpha$-pinene and cyclohexene against the different compositions. The results for both VOC imply that a high oxygen number is

the most essential parameter for lowered volatility. Thus, the ability to undergo efficient autoxidation may be more important for SOA formation than the size of the initial VOC reactant.

*Data availability.* Data will be available at https://doi.org/10.5281/zenodo.4001577

*Author contributions.* The study was conceived by ME, MRiv and MRis. The measurements were performed by MRiv, OG and LQ. MRä did the main data analysis and visualization under the supervision of ME and with additional guidance from OP and MRis. MRä wrote the

manuscript, and it was reviewed and commented by the other authors.

*Competing interests.* The authors declare that they have no conflict of interest.

*Acknowledgements.* This work was supported by King Abdullah University of Science and Technology (Award OSR-2016-CRG5-3022), the European Research Council (Grant 638703-COALA), and the Academy of Finland Centre of Excellence (Grants 307331, 317380, 320094 and 326948). We thank Liine Heikkinen for assistance during the experiments, Pontus Roldin for helpful discussion, and the tofTools team

for providing tools for mass spectrometry data analysis.



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
