# Peer review of "Effects of $NO_x$ and seed aerosol on highly oxygenated organic molecules (HOM) from cyclohexene ozonolysis"

_Atmospheric Chemistry and Physics, 2020_

## Referee Comment (RC1) · Anonymous Referee #1 · 15 Oct 2020

This manuscript presents gas-phase measurements of highly oxygenated organic molecules (HOMs) derived from cyclohexene ozonolysis, made using a nitrate chemical ionization mass spectrometer (NO3-CIMS), as a function of initial NOx mixing ratio and ammonium bisulfate (ABS) seed concentration. Comparison of mass spectra for experiments with and without NO addition reveals a general decrease in the abundance of C11–12H16–22O7–16 dimers, varied responses for C3–6H4–12O3–10 monomers, and increases in both N-containing monomers (C5–6H7–11NO4–10) and dimers (C11–12H17–20NO9–14). Based on the observed uptake of compounds to the ABS seed, the authors propose that condensation is primarily controlled by the number of oxygen atoms present in each molecule. The measured gas-phase losses of

monomers, dimers, and N-containing compounds due to condensation are also compared to those predicted by two models designed to estimate volatilities of atmospheric oxidation products.

In many respects, this work is a replication of the recent study by Peräkylä et al. (2020), focusing on HOMs derived from ozonolysis of cyclohexene rather than a-pinene. Atmospheric interest in cyclohexene ozonolysis is as a surrogate for a-pinene, given the potential to derive generalizable mechanistic insight from its comparatively simpler symmetric structure. Although this manuscript has the potential to complement and extend the findings of Peräkylä et al. (2020), in its current form it presents few novel results and the discussion is lacking. In particular, molecular-level interpretation of the observed effects of NOx on the abundance and distribution of cyclohexene-derived HOMs is limited and often overlooks key findings from recent papers by co-author Matti Rissanen on HOMs formation in the cyclohexene ozonolysis system. For these reasons, I recommend that publication be considered only after the comments detailed below are addressed.

Specific Comments

1. Line 207: Dependence of $C_6H_8O_7$ and $C_6H_8O_9$ on NOx. Given that Rissanen et al. (2014) proposes structures and formation mechanisms for these HOMs based on quantum chemical calculations and experiments with isotopic labeling, "speculating the reason for this difference is difficult..." is a rather unsatisfying interpretation. A discussion that specifically addresses whether the observed NOx trends are consistent with the Rissanen et al. (2014) structures/mechanisms would strengthen this section considerably. Notably, Rissanen et al. (2014) states that "it seems more likely that $C_6H_8O_7$ product is formed through bimolecular reactions of the intermediate peroxy radials (i.e., by RO2 + RO2 reactions)," which is consistent with the observed decrease in the $C_6H_8O_7$ signal with increasing NO. Additionally, Rissanen et al. (2018) states that "at a low addition level, NO aids the $C_6H_8O_8$ HOM product formation by reactive alkoxy radical (RO) formation and illustrates the oxidation enhancing influence of generating highly reactive RO radicals." A similar role of RO reactions in the formation of the C6H8O9 HOM may account for the observed increase in signal with increasing NO.

2. Line 184: Role of NO2. Despite the statement that "only NO mainly has an impact on the radical chemistry," NO2 should also exert a considerable influence in this system via reaction with acylperoxy radicals [RC(O)O2]. Rissanen et al. (2014) proposes that acylperoxy radicals (C6H9O6 and C6H9O8) are key intermediates in the formation of the major C6H8O7 and C6H8O9 HOMs, while Rissanen et al. (2018) implicates acylperoxy radicals in the production of C12 dimers: "these observations imply the special importance of acylperoxy radicals in directing autoxidation phenomena." Given the reported NO/NOx ratio of ∼3%, the role of NO2 should be accounted for.

3. Kinetic Modeling. Interpretation of the observed NOx dependences would greatly benefit from a kinetic box model, parameterized with known rate constants where available, calculated rate constants from Rissanen et al. (2014), and structure-activity relationships from Jenkin et al. (2019). For example, as a function of the initial NO mixing ratio: What fraction of O3 reacts with C6H10 vs. NO? What fraction of the initially formed C6H9O4-RO2 undergoes unimolecular isomerization vs. reaction with NO or other RO2? How much NO3 is formed? What fraction of the C6H10 decay is due to reaction with NO3 vs. O3 vs. OH? What fraction of OH reacts with NO2 vs. C6H10?

4. Figures 2 and 3. Molecular formulas should either be rotated 90 degrees and placed directly above the corresponding peaks or moved into the white space at the top of the figure and linked to peaks with arrows. The existing format is cluttered and difficult to interpret. Also, the y-axis should be labeled "normalized ion counts," given the normalization to the reagent ion signals.

5. Figure 4. The discussion of Figure 4 is sparse and does not provide any information beyond that presented in relation to Figure 5. However, there are some interesting non-linear NOx dependences displayed in Figure 4. For example, the C12H19NO11 signal decreases between 8 and 10 ppb NOx after increasing from 1 to 4 ppb NOx, presum-

ably due to suppression of RO2 + RO2 at sufficiently high NO. Additionally, the traces for C6H8O7 and C12H20O9 are effectively superposable, suggesting similar formation pathways. The discussion of this figure should be expanded to include interpretation of these trends.

6. Figures 5 and 6. The use of a color scale in Figure 5 that incorporates opacity makes it very difficult to determine if the degree of marker transparency for a given compound is due to the fit factor, relative signal change, or a combination of the two. Instead of transparency, perhaps different marker symbols (e.g., square, triangle, circle, diamond) could be used to denote different fit factor ranges (e.g., <0, 0.1-0.4, 0.5-0.7, 0.8-1.0). For consistency, the use of symbols to indicate fit factor ranges should also be applied to Figure 6.

7. Line 223: Dimers from O3- and NO3-Derived RO2. The proposed formation of N-containing dimers via cross-reactions of O3- and NO3-derived RO2 is quite interesting and possibly the first account of such chemistry. Several recent studies (Zhao et al. 2018, Kenseth et al. 2018, and Li et al. 2019) report on the analogous formation of dimers from cross-reactions of O3- and OH-derived RO2. A discussion of the current work as it relates to these studies and the potential importance of multi-oxidant chemistry would be useful.

8. Line 259: Model Adaptation. Rather than shifting the fit between FR and log10(C*) reported for a-pinene in Peräkylä et al. (2020) by one unit to account for the difference in SOA concentration, which seems somewhat arbitrary, why not generate a new logistic fit using values for cyclohexene model compounds calculated by the ADCHAM model?

9. Line 282: Model-Measurement Agreement. Both models significantly overpredict the FR for compounds between m/z 250 and 300, including the major C6H8O7 and C6H8O9 HOMs. Was the FR overpredicted for all observed HOMs, suggesting an inability of the models to effectively capture HOMs volatilities? A more detailed discussion of the potential causes of the model-measurement disagreement (e.g., ability to accurately model contributions of key HOMs functionalities, such as hydroperoxides, to volatility) would be beneficial.

10. Line 292: Scaling Experimental Data. What assumptions were made in scaling the data from Peräkylä et al. (2020) to facilitate comparison with the results obtained in this work under different experimental conditions? Such a comparison seems somewhat contrived and the utility is not immediately clear, however, beyond stating that the scaled data are included in Figure 9 they are not discussed. The authors should either include a comparison of the elemental composition vs. volatility trends for cyclohexene- and a-pinene-derived HOMs or remove the Peräkylä et al. (2020) data from Figure 9. As an example, Figure 6 in Peräkylä et al. (2020) suggests that m/z > 300 is required for the FR of non-N-containing a-pinene compounds to drop below 50%, whereas Figure 8 in this work shows that for non-N-containing cyclohexene compounds m/z > 250 corresponds to FR values below 50%. A discussion of such differences would be informative.

Minor Comments

1. Line 29. Sentence beginning with "Through oxidation..." is awkward. Consider rephrasing.

2. Line 76. What is the detection limit of the NOx analyzer?

3. Line 81. Sentence beginning with "Experiments with ammonium sulfate..." is unnecessary.

4. Figure 7. Why is there so much variability in the signal of the three species prior to ABS seed injection ($\sim$0.9-1.1)? What are the uncertainties in the measured signals of the detected species and their relative changes as a function of NOx mixing ratio and ABS seed concentration?

5. Line 244. How were the modeled values "scaled to the range of observed remaining

fractions"?

6. Line 250. A discussion of the suitability of using an activity coefficient of unity and the potential impacts of this assumption should be included.

7. Line 315. This sentence is contradictory. By definition, "increased termination of RO2 radicals by NO" does not lead "to more alkoxy radicals" but rather to closed-shell alkyl nitrates. Radical propagation via reaction of RO2 with NO will produce more RO radicals, which seems to be the intent of the sentence.

References

Jenkin, M. E., Valorso, R., Aumont, B., Rickard, A. R.: Estimation of rate coefficients and branching ratios for reactions of organic peroxy radicals for use in automated mechanism construction, Atmos. Chem. Phys., 19, 7691–7717, 2019.

Kenseth, C. M., Huang, Y., Zhao, R., Dalleska, N. F., Hethcox, J. C., Stoltz, B. M., Seinfeld, J. H.: Synergistic O3 + OH oxidation pathway to extremely low-volatility dimers revealed in b-pinene secondary organic aerosol, Proc. Natl. Acad. Sci. U.S.A., 115, 8301–8306.

Li, X., Chee, S., Hao, J., Abbatt, J. P. D., Jiang, J., Smith, J. N.: Relative humidity effect on the formation of highly oxidized molecules and new particles during monoterpene oxidation, Atmos. Chem. Phys., 19, 1555–1570, 2019.

Peräkylä, O., Riva, M., Heikkinen, L., Quéléver, L., Roldin, P., and Ehn, M.: Experimental investigation into the volatilities of highly oxygenated organic molecules (HOMs), Atmos. Chem. Phys., 20, 649–669, 2020.

Rissanen, M. P.: NO2 Suppression of Autoxidation – Inhibition of Gas-Phase Highly Oxidized Dimer Product Formation, ACS Earth Space Chem., 2, 1211–1219, 2018.

Rissanen, M. P., Kurten, T., Sipilä, M., Thornton, J. A., Kangasluoma, J., Sarnela, N., Junninen, H., Jorgensen, S., Schallhart, S., Kajos, M. K., Taipale, R., Springer, M.,

Mentel, T. F., Ruuskanen, T., Petäjä, T., Worsnop, D. R., Kjærgaard, H. G., and Ehn, M.: The Formation of Highly Oxidized Multifunctional Products in the Ozonolysis of Cyclohexene, J Am. Chem. Soc., 136, 15596–15606, 2014.

Zhao, Y., Thornton, J. A., Pye, H. O. T.: Quantitative constraints on autoxidation and dimer formation from direct probing of monoterpene-derived peroxy radical chemistry, Proc. Natl. Acad. Sci. U.S.A., 115, 12142–12147.

---

## Referee Comment (RC2) · Anonymous Referee #2 · 22 Oct 2020

Raty et al. generated highly oxygenated organic molecules (HOM) from the ozonolysis of cyclohexene in an environmental chamber. They separately characterized the effects of adding NO and ammonium bisulfate (ABS) seed particles on cyclohexene HOM composition.  HOM were detected with a time-of-flight chemical ionization mass spectrometer using nitrate reagent ion. Following NO addition to the chamber, the relative abundance of $C_6H_8O_9$, $C_6H_9NO_9$, $C_{12}H_{19}NO_{11}$ (and other HOM) increased, especially nitrogen-containing HOM. The abundance of $C_6H_8O_7$, $C_{12}H_{20}O_9$ and other HOM decreased. Following ABS addition to the chamber, signals at $C_6H_{10}O_4$, $C_6H_{10}O_6$, and $C_6H_{10}O_8$ – and other low-volatility HOM - decreased due to the increased condensation sink. A model was used to relate the fraction of HOM remaining in the gas phase to its effective saturation concentration ($C^*$). Overall, the experiments are well motivated from the perspective of trying to better understand (1) the composition of molecules that contribute to new particle formation and (2) the effects of $NO_x$ and condensation sink perturbations on NPF. However, in my opinion, the novelty and atmospheric significance of the results are uncertain the way they are currently presented. The comments below should be implemented into a revised manuscript before I would support eventual publication in ACP.

**General Comments**

1. Peräkylä et al. (2020) describe a similar set of experiments with a different precursor ($\alpha$-pinene). In that study, in general, I felt that the analysis was clearer and more thorough than was presented here. At the very least, because a study of cyclohexene-derived HOM is motivated here as "a surrogate for monoterpenes with an endocyclic double bond" to assess "how applicable earlier volatility parameterisations are on the cyclohexene system" a revised manuscript should incorporate parallel analyses to what were presented in the companion Peräkylä et al. (2020) manuscript. For example, Figure 8 in Peräkylä et al. (2020) shows the calculated $C^*$ values of $C_{10}H_{16}O_x$ HOM that were calculated from seed perturbation experiments. Actual $C^*$ values of cyclohexene-derived HOM are never plotted or discussed here as far as I can tell. Also, Figure 7 of Peräkylä et al. (2020) shows a scatter plot comparing the modeled versus measured fraction remaining (FR) of a-pinene-derived HOM. In my opinion this is a much clearer presentation than Figure 8 in this manuscript.

2. After setting up Section 2.3 for a discussion of the cyclohexene ozonolysis mechanism, it transitions to a more abstract/general discussion after L136. I think it would be better to focus the discussion on what happens to the $C_6H_9O_4$ peroxy radical, such as the specific autooxidation and/or $RO_2$-$RO_2$ reactions that lead to some of the major HOM products, i.e. $C_6H_8O_7$, $C_6H_8O_9$, and $C_{12}H_{20}O_9$, which have already been identified in earlier studies (e.g. Rissanen et al.). A figure with a reaction scheme showing these autooxidation steps would also be useful. Reframing the discussion around specific autooxidation steps that lead from $C_6H_9O_4$  to HOM, along with a reaction scheme, allows for a more direct transition to the results and discussion of the $NO_x$ and condensation sink perturbation studies.

3. The way the paper is currently written, the relative roles of $RO_2$ + NO and $RO_2$ + $NO_3$ reactions in generating the results that are presented/discussed in Figure 3 and Section 3.1 are not clear: [NO] is below detection limit, and [NO3] is not constrained by measurements and/or modeling. At the least, a photochemical box model simulation (e.g. KinSim or similar, see Peng and Jimenez, 2019) with the relevant COALA chamber conditions, reactions and rate coefficients would be

appropriate here, perhaps as an appendix. Because the use of cyclohexene is motivated as a simple surrogate for monoterpenes, it would also be appropriate to add another reaction scheme to Section 3.1 that explains the increases or decreases in HOM observed in Figures 3-4 following perturbation by NO (and/or $NO_3$).

**Minor/Technical Comments**

4. **L67**: The authors state: "The resulting steady state concentration of ozone was approximately 18 ppb, while a rough estimate for cyclohexene concentration was about 100 ppb […] estimated from the difference in ozone concentration with and without cyclohexene". What was the ozone concentration prior to cyclohexene addition? This would be useful for any readers that might try to reproduce the experimental conditions described here. To what extent is the HOM composition, e.g. the monomer:dimer ratio, influenced by [cyclohexene]:[ozone]?

5. **L77**: Please explain why 9 ug/m$^3$ loading of ABS, with corresponding condensation sink of ~0.085 s$^{-1}$, was chosen for the seed perturbation studies. Additionally, ozonolysis of ~100 ppb cyclohexene presumably generates some SOA given reported SOA mass yields of approximately 0.15 – 0.20 (Keywood et al., 2004). If that's the case here, what is the condensation sink of homogenously nucleated cyclohexene ozonolysis SOA relative to the added ABS seeds?

6. Figure 1 and some of the accompanying discussion could probably be moved to an appendix or supplement.

7. Figure 2 and 3 could be combined into a single 2-panel figure to facilitate easier comparison. I would also consider simply adding a 3$^{rd}$ panel showing the mass spectrum of cyclohexene HOM following the addition of 0.085 s$^{-1}$ ABS condensation sink, and removing Figure 7. Unlike Figure 4, which shows the change in HOM across a continuum of $NO_x$ values, there is only one ABS condensation sink value, so there are no meaningful trends to show in Figure 7 that couldn't be more simply presented as a mass spectrum to directly compare with Figure 2.

8. Figure 9: I get what the authors are trying to do here, but I find this figure very difficult to read, and as the authors note, quantitatively comparing FR values of cyclohexene and $\alpha$-pinene HOM is not straightforward because of different SOA loadings and corresponding condensation sinks in the different studies. I suggest removing the $\alpha$-pinene HOM from this figure, and then adding a separate figure plotting C* values of $C_6H_{10}O_x$, $C_6H_{10}NO_x$, $C_{10}H_{16}O_x$ and $C_{10}H_{16}NO_x$ HOM as calculated from the seed perturbation studies.

**References**

M. D. Keywood, J. H. Kroll, V. Varutbangkul, R. Bahreini, R. C. Flagan, and J. H. Seinfeld. Secondary organic aerosol formation from cyclohexene ozonolysis: effect of OH scavenger and the role of radical chemistry. Environ Sci Technol. 2004;38(12):3343-50. doi: 10.1021/es049725j.

Zhe Peng and Jose L. Jimenez. KinSim: A Research-Grade, User-Friendly, Visual Kinetics Simulator for Chemical-Kinetics and Environmental-Chemistry Teaching, *Journal of Chemical Education* 2019; *96* (4), 806-811, DOI: 10.1021/acs.jchemed.9b00033.

---

## Author Comment (AC1) · 21 Feb 2021

**Authors' response to referee comments**

We thank the referees for these comments. We provide our point-by-point responses below.

**Referee #1**

This manuscript presents gas-phase measurements of highly oxygenated organic molecules (HOMs) derived from cyclohexene ozonolysis, made using a nitrate chemical ionization mass spectrometer (NO3-CIMS), as a function of initial NOx mixing ratio and ammonium bisulfate (ABS) seed concentration. Comparison of mass spectra for experiments with and without NO addition reveals a general decrease in the abundance of C11– 12H16–22O7–16 dimers, varied responses for C3–6H4–12O3–10 monomers, and increases in both N-containing monomers (C5–6H7–11NO4–10) and dimers (C11–12H17–20NO9–14). Based on the observed uptake of compounds to the ABS seed, the authors propose that condensation is primarily controlled by the number of oxygen atoms present in each molecule. The measured gas-phase losses of monomers, dimers, and N-containing compounds due to condensation are also compared to those predicted by two models designed to estimate volatilities of atmospheric oxidation products.

In many respects, this work is a replication of the recent study by Peräkylä et al. (2020), focusing on HOMs derived from ozonolysis of cyclohexene rather than a-pinene. Atmospheric interest in cyclohexene ozonolysis is as a surrogate for a-pinene, given the potential to derive generalizable mechanistic insight from its comparatively simpler symmetric structure. Although this manuscript has the potential to complement and extend the findings of Peräkylä et al. (2020), in its current form it presents few novel results and the discussion is lacking. In particular, molecular-level interpretation of the observed effects of NOx on the abundance and distribution of cyclohexene-derived HOMs is limited and often overlooks key findings from recent papers by co-author Matti Rissanen on HOMs formation in the cyclohexene ozonolysis system. For these reasons, I recommend that publication be considered only after the comments detailed below are addressed.

R1: As a general remark concerning the referee's comment about molecular-level interpretations, these types of interpretations are indeed quite limited in our manuscript. However, this was to a large extent a deliberate choice, as our approach is quite different from that of e.g. Rissanen et al. (2014) or Rissanen (2018). The underlying challenge is that the high-resolution mass spectrometers used in all these studies produce an enormous amount of different signals. Even for relatively simple systems, the amount of identifiable peaks are in the hundreds, often nearing a thousand (Riva et al., 2019). For the analysis in our manuscript, we fitted 240 different ions to each of the measured mass spectra. This inevitably means that one must choose the level of molecular detail that a study will cover. In the two papers by Rissanen et al., the focus was on detailed mechanisms, and in practice only around five specific ions in each paper were used for the main analysis. This is in sharp contrast to our manuscript, where we attempt to include as many signals as possible from the mass spectrum in order to be comprehensive and maximize the amount of signal that can be used. This, however, comes at the obvious expense of molecular detail.

Another way of comparing the approaches is that in Rissanen (2018), only one of the figures contains measured data, whereas in this manuscript all 9 figures present experimental data. We believe that both these types of approaches are needed, and we further hope that the methods we have used here, e.g. the FitFactor, would facilitate more of the types of studies that we have done here, which utilize the entire mass spectral information that the mass spectrometers provide.

Nevertheless, we have concluded that the manuscript type "Measurement report" is more appropriate for our work, and the intended scope of manuscript. We hope that this will make it less likely for readers to have wrong expectations concerning the content of the study.

1. Line 207: Dependence of C6H8O7 and C6H8O9 on NOx. Given that Rissanen et al. (2014) proposes structures and formation mechanisms for these HOMs based on quantum chemical calculations and experiments with isotopic labeling, "speculating the reason for this difference is difficult..." is a rather unsatisfying interpretation. A discussion that specifically addresses whether the observed NOx trends are consistent with the Rissanen et al. (2014) structures/mechanisms would strengthen this section considerably. Notably, Rissanen et al. (2014) states that "it seems more likely that C6H8O7 product is formed through bimolecular reactions of the intermediate peroxy radials (i.e., by RO2 + RO2 reactions)," which is consistent with the observed decrease in the C6H8O7 signal with increasing NO. Additionally, Rissanen et al. (2018) states that "at a low addition level, NO aids the C6H8O8 HOM product formation by reactive alkoxy radical (RO) formation and illustrates the oxidation enhancing influence of generating highly reactive RO radicals." A similar role of RO reactions in the formation of the C6H8O9 HOM may account for the observed increase in signal with increasing NO.

R2: In reference to R1, we have avoided going too much into detail for any specific molecules. However, we have now extended the discussion on this section, also better referencing these earlier studies so that readers can more easily find further information.

We will also point out that, slightly counter intuitively, some RO2 lifetimes can in fact increase with addition of NO, and therefore the increased concentrations of products do not automatically mean that alkoxy radicals are involved, as one could assume from the above quote from Rissanen et al. (2018). For example, Berndt et al. (2020) showed that HOM RO2 can have reaction rate coefficients on the order of 10^-10 cm^s^-1 with other RO2, while self-reactions of the less oxygenated, but more abundant, RO2 are closer to 10^-12. In this case, if all of the RO2 would react at rates of 10^-11 cm^3 s^-1 with NO, then at low NO additions the HOM RO2 sink would actually decrease as the less oxygenated HOM decrease. In addition, increased NO + HO2 reactions will lead to increased OH concentrations, which can impact the product distributions.

Given all these possible effects, we do not feel like including this type of detailed speculation would improve our manuscript, as many more types of conditions would need to be probed in order to distinguish between those different pathways.

**Modified section in manuscript:**

"This varied response in the monomer signals highlights how complicated the formation pathways can be, even when the initial reactant is a relatively simple molecule like cyclohexene. Speculating on the exact reasons for differences between individual signal responses is difficult. A single observed elemental composition may often contain a collection of isomers, each with distinct formation pathways. The reaction rate constants and different branching ratios of reactions between different combinations of RO2 are also unknown, and can vary by orders of magnitude depending on the specific structures (Shallcross et al., 2005; Berndt et al., 2018). Therefore, any change in the reactant concentration, timescales, or other conditions, can yield very different results, with highly non-linear responses, making direct comparisons with other studies challenging. For these reasons, we will only briefly look at certain example molecules and compound groups in more detail. The reason for the decreasing  $C_6H_8O_7$  could be a decrease in bimolecular reactions of intermediate peroxyradicals. Rissanen et al. (2014) have previously suggested their likely formation pathway to involve a termination to a closed cell molecule in a  $RO_2+RO_2$  reaction. With the addition of NO, these types of reactions should decrease, and our results indeed show such a decrease (Fig. 3). The concurrent rising trend of  $C_6H_8O_9$  suggests that its formation pathway may involve alkoxy radical intermediates, which can form both from  $RO_2+RO_2$  (R6) and  $RO_2+NO$  (R7) reactions. Similar to our results, (Rissanen, 2018) saw an increase of  $C_6H_8O_8$  with NO, and suggested the involvement of alkoxy radicals. Overall, we noticed increasing trends for at least 20 different monomer compounds that did not contain N-atoms. As a notable exception, most of the  $C_6H_10O_x$  compounds showed a decrease, which may be explained by a decrease in the bimolecular reactions (R3) and (R5). Most of the less oxidised SVOC only showed relatively small changes. Finally,  $NO_x$  will also function as a sink for OH, although cyclohexene should still dominate the OH reactivity in the chamber even at the highest  $NO_x$  concentration."

2. Line 184: Role of NO2. Despite the statement that "only NO mainly has an impact on the radical chemistry," NO2 should also exert a considerable influence in this system via reaction with acylperoxy radicals [RC(O)O2]. Rissanen et al. (2014) proposes that acylperoxy radicals (C6H9O6 and C6H9O8) are key intermediates in the formation of the major C6H8O7 and C6H8O9 HOMs, while Rissanen et al. (2018) implicates acylperoxy radicals in the production of C12 dimers: "these observations imply the special importance of acylperoxy radicals in directing autoxidation phenomena." Given the reported NO/NOx ratio of \_3%, the role of NO2 should be accounted for.

R3: We reformulated this section to be more clear that also NO2 will be playing some role in our chamber. NO3 is also important, as can be seen from the N-containing dimer signals. However, we had also noted in the manuscript that we do not attempt to distinguish between the roles of NO, NO2 or NO3, as a given elemental composition can have contributions from all of these pathways, and mass spectrometry alone cannot distinguish between these.

We have now expanded on this discussion as written below:

"Injecting NO into the chamber has a significant impact on the chemistry within, which consequently leads to changed product signals. Although we expect NO to have the largest effect on the radical chemistry (Eqs. R7-R9), due to the aforementioned technical limitations leading to a lack of separate measurements of NO, we compare signals to the total measured  $NO_x$ .  $NO_2$  can also have some noticeable effect, by reactions with acylperoxy radicals (Eq. R9), which are expected to be abundant in this system (Rissanen et al., 2014; Rissanen, 2018), but as discussed in section 2.3, we do not expect this to be a significant process in our system. In addition, also oxidation by  $NO_3$  radicals will take place in the chamber. As our observations are limited to the elemental composition of the products, and we have no additional information on specific structures, we do not attempt to distinguish between the effects of NO,  $NO_2$ , and  $NO_3$ ."

3. Kinetic Modeling. Interpretation of the observed NOx dependences would greatly benefit from a kinetic box model, parameterized with known rate constants where available, calculated rate constants from Rissanen et al. (2014), and structure-activity relationships from Jenkin et al. (2019). For example, as a function of the initial NO mixing ratio: What fraction of O3 reacts with C6H10 vs. NO? What fraction of the initially formed C6H9O4-RO2 undergoes unimolecular isomerization vs. reaction with NO or other RO2? How much NO3 is formed? What fraction of the C6H10 decay is due to reaction with NO3 vs. O3 vs. OH? What fraction of OH reacts with NO2 vs. C6H10?

R4: We agree that the type of kinetic modelling suggested by the reviewer would be very interesting. Still, we feel it is out of scope for our manuscript, with the foremost reasons outlined here:

- There are simply too many unknowns and uncertainties. For example, due to instrumental challenges, both the exact amount of cyclohexene and NO have unfortunately large uncertainties in our study. Also the rate constants for H-shifts from Rissanen et al. (2014) carry with them large uncertainties, and these would need to compete with reactions with NO, NO2, NO3 and other RO2 at rates we would need to guess. And as shown by Berndt et al. (2018), these rates are very sensitive to the exact radical structures.
- 2. As noted in R1, our aim was to provide a comprehensive analysis of the observed molecules, not to target a detailed understanding of a subset of molecules. The suggested kinetic modelling would only be able to target a few "known" intermediates.
- 3. With the current focus of our paper, even with all the answers to the questions raised by the referee, our current conclusions would not be much impacted. As such, including the suggested modelling, although certainly interesting, would feel disproportionately laborious.

Therefore, we would not attempt such a modelling study. However, we hope that the comprehensive data we present here would be utilized in a more dedicated modelling study in the future.

4. Figures 2 and 3. Molecular formulas should either be rotated 90 degrees and placed directly above the corresponding peaks or moved into the white space at the top of the figure and linked to peaks with arrows. The existing format is cluttered and difficult to interpret. Also, the y-axis should be labeled "normalized ion counts," given the normalization to the reagent ion signals.

R5: We thank the referee for this suggestion. The formulas have now been rotated 90 degrees, and we have also combined the two plots into one figure.

The y-axis shows the absolute (non-normalised) ion count, so the label was correct. As these are merely examples of spectra with the biggest peaks shown, we wanted to leave the counts as the unit, as it gives one an idea of the statistics of our data.

5. Figure 4. The discussion of Figure 4 is sparse and does not provide any information beyond that presented in relation to Figure 5. However, there are some interesting nonlinear NOx dependences displayed in Figure 4. For example, the C12H19NO11 signal decreases between 8 and 10 ppb NOx after increasing from 1 to 4 ppb NOx, presumably due to suppression of RO2 + RO2 at sufficiently high NO. Additionally, the traces for C6H8O7 and C12H20O9 are effectively superposable, suggesting similar formation pathways. The discussion of this figure should be expanded to include interpretation of these trends.

R6: The main idea of figure 4 is to act as an introduction to figure 5 by showing a few examples of how the data look. Figure 5 has a lot of information, and we feel like this extra step may help readers to follow along better. However, we acknowledge that this figure would benefit from some additional discussion, which we have added to the section:

"This figure also exemplifies the possible non-linear effects of  $NO_x$  on different molecules. While some molecules do show a linear trend with the  $NO_x$  addition, the signal of the nitrate dimer  $C_{12}H_{19}NO_{11}$  for example, initially increases with the added  $NO_x$ , as oxidation by  $NO_3$  generates nitrogen-containing radicals, allowing for the formation of such a dimer. However, when  $NO_x$  concentration increases (>8 ppb), the suppression of  $RO_2+RO_2$  reactions by NO (and  $NO_2$ ) starts to become more important and we observe a decreasing trend."

6. Figures 5 and 6. The use of a color scale in Figure 5 that incorporates opacity makes it very difficult to determine if the degree of marker transparency for a given compound is due to the fit factor, relative signal change, or a combination of the two. Instead of transparency, perhaps different marker symbols (e.g., square, triangle, circle, diamond) could be used to denote different fit factor ranges (e.g., <0, 0.1-0.4, 0.5-0.7, 0.8-1.0). For consistency, the use of symbols to indicate fit factor ranges should also be applied to Figure 6.

R7: We noticed that figure 5 included colours outside the shown colour scale (for data points outside the shown limits). This allowed fairly dark blue markers in the figure, which made it harder to distinguish between transparency and relative change. This has now been corrected, which hopefully also helps with the problem mentioned by the referee, as darker markers should always indicate a lower fitFactor.

Interpreting the exact colours of very low FitFactor signal markers will be difficult, but this is also intentional. These signal changes are associated with very high uncertainties, and we therefore downplay their "visual contribution" to the plot. Signal changes of compounds with high concentrations and high FitFactors are the most reliable signals, and are by design the ones that stand out most easily.

We also considered the option of using different marker symbols. However, in addition to not having the effect of diminishing the impact of uncertain signals, we feel like this will make interpreting relative signal sizes (=marker size) more difficult, while opacity allows an easier and quicker way to identify the more reliable signals which one should focus on, in comparison to having to distinguish between different symbols.

Figure 5 is overall challenging as it presents six different parameters for each data point plotted, namely number of C, H, and O, as well as FitFactor, concentration, and relative change. Still, we feel that the main message does come across in the updated figure. We also added a second colour bar, corresponding to an opacity of 0.5, which should provide additional help in the interpretation. In addition, the colour bars now have the same grey background colour as in the plot, which makes distinguishing between the colours easier. We also added a sentence about "outlier values" with high FitFactors to the caption.

7. Line 223: Dimers from O3- and NO3-Derived RO2. The proposed formation of N-containing dimers via cross-reactions of O3- and NO3-derived RO2 is quite interesting and possibly the first account of such chemistry. Several recent studies (Zhao et al. 2018, Kenseth et al. 2018, and Li et al. 2019) report on the analogous formation of dimers from cross-reactions of O3- and OH-derived RO2. A discussion of the current work as it relates to these studies and the potential importance of multi-oxidant chemistry would be useful.

R8: Cross-reactions of O3- and NO3 –derived radicals forming N-containing dimers have been reported on earlier both in the atmosphere (Yan et al., 2016; Jokinen et al., 2017; Zha et al., 2018; Zhang et al., 2020) and in the laboratory (Yan et al., 2020). In particular, both Yan et al. (2016) (reactions 10-12 in the paper) and Zhang et al. (2020) explicitly discuss the nitrate dimers as products of cross reactions between O3- and NO3-derived RO2. We added additional references to these studies, but prefer not to go into detailed discussions on dimer formation pathways from different oxidants, for reasons discussed above and in R1.

The relevant discussion now with the added references:

"Nitrogen-containing dimers are also detected. This implies that some of the cyclohexene had been oxidised by  $NO_3$ , which had led to the formation of nitrogen containing radicals. These radicals then likely formed dimers in reactions with ozone-derived  $RO_2$ , as the dimers only had one nitrate functionality. Similar nitrogen-containing HOM dimers have been observed also in field measurements where monoterpenes were the dominant VOC (Zha et al., 2018; Jokinen et al., 2017), and in studies by Yan et al. (2016) and Zhang et al. (2020) they have been directly implicated as cross reaction products of  $RO_2$  formed in  $NO_3$  and  $O_3$  oxidation."

8. Line 259: Model Adaptation. Rather than shifting the fit between FR and log10(C\*) reported for a-pinene in Peräkylä et al. (2020) by one unit to account for the difference in SOA concentration, which seems somewhat arbitrary, why not generate a new logistic fit using values for cyclohexene model compounds calculated by the ADCHAM model?

R9: We would not consider the shift "arbitrary", as an order of magnitude increase in aerosol mass should shift the curve by close to one order of magnitude based on equilibrium partioning theory (Donahue et al. 2012). The alternative suggested by the referee, to utilize ADCHAM and cyclohexene model compounds, would also involve notable uncertainties due to uncertainties in vapour pressures of relevant models of highly oxidized compounds.

We acknowledge that our approach is not perfect, but as stated in the manuscript, for the comparison that the data are used for, we believe our approach is more than sufficient. However, we realized that we had not included the reference behind this shift, so we modified the sentence:

"By following equilibrium partitioning theory (Donahue et al., 2012), to account for the difference, we shifted the original fit between FR and  $\log_{10}(C^*)$  left by 1 (i.e. plugging in  $x = \log_{10}(C^*)+1$ ), thus adjusting the relationship between FR and C\* by a factor of 10."

9. Line 282: Model-Measurement Agreement. Both models significantly overpredict the FR for compounds between m/z 250 and 300, including the major C6H8O7 and C6H8O9 HOMs. Was the FR overpredicted for all observed HOMs, suggesting an inability of the models to effectively capture HOMs volatilities? A more detailed discussion of the potential causes of the model-measurement disagreement (e.g., ability to accurately model contributions of key HOMs functionalities, such as hydroperoxides, to volatility) would be beneficial.

R10: This type of discussion was indeed missing, and we have now added more. Following comments from both referees concerning the original Fig. 9 (now Figure 7), we made changes to it as well, and most of the requested discussion is added to the section concerning Fig. 9. This felt more natural, since at that stage we have shown that the surprisingly efficient condensation/uptake is primarily driven by the oxygen content in the HOM.

This expanded discussion begins like so:

"Both the Peräkylä et al. (2020) and Bianchi et al. (2019) models successfully predict the high volatility of the least oxidised monomers, and the irreversible condensation of the largest dimers. They locate the steepest transition in volatility to the region between monomers and dimers, and in disagreement with our observations, fail to capture the observed nearly irreversible condensation of many of the more oxygenated monomers as well. The Bianchi et al. (2019) model predicts the irreversible condensation of all of the dimers, but the decrease in volatility in the Peräkylä et al. (2020) model is a more gradual function of mass and therefore predicts a condensation weaker than observed. The discrepancy between the observations and modelled FR is also clearly visible in Figure B1.

As the models misplace the transition between non-condensing and irreversibly condensing compounds, they also overpredict the product volatilities in general. Since we cannot distinguish the differences in volatilities of compounds that condensed irreversibly in our experiments, we are not able to confirm where the observed volatilities merge with the ones predicted by the models.

In order to better understand what controls the volatility of cyclohexene products and where these models might fail, we had a more detailed look into the effect of elemental composition on FR..."

10. Line 292: Scaling Experimental Data. What assumptions were made in scaling the data from Peräkylä et al. (2020) to facilitate comparison with the results obtained in this work under different experimental conditions? Such a comparison seems somewhat contrived and the utility is not immediately clear, however, beyond stating that the scaled data are included in Figure 9 they are not discussed. The authors should either include a comparison of the elemental composition vs. volatility trends for cyclohexene and a-pinene-derived HOMs or remove the Peräkylä et al. (2020) data from Figure 9. As an example, Figure 6 in Peräkylä et al. (2020) suggests that m/z > 300 is required for the FR of non-N-containing a-pinene compounds to drop below 50%, whereas Figure 8 in this work shows that for non-N-containing cyclohexene compounds m/z > 250 corresponds to FR values below 50%. A discussion of such differences would be informative.

R11: (Here we simultaneously also give our response to the second referee's comment 8.) The referees are correct that the a-pinene data in Figure 9 was not utilized very much, and it caused the main message of the figure to become blurred. We removed the data from the plots, as suggested. We have instead expanded on the discussion around the observed behaviour, and connected it more clearly with the discussion concerning Figure 8 (/now figure 6) and the observed model-observation discrepancies. We have also added additional figures to an appendix (e.g. see R16).

The expanded discussion continues as follows: (former Fig. 9 is now Fig. 7)

"...In Figure 7, the fraction remaining is plotted separately against carbon, hydrogen and oxygen numbers, as well as the "effective O:C ratio"  $(n_0-2n_N)/n_c$ . The incentive for choosing this ratio, rather than simply

O:C, is the same as for the modification of the Bianchi et al. (2019) model, i.e. to account for nitrate functionalities. For non-nitrates this expression reduces to an O:C ratio.

Cyclohexene oxidation products with the lowest carbon and hydrogen content hardly condensed, while the opposite was true for the molecules with the highest C and H content, mostly associated with the dimers (Fig. 7a and 7b). However, in the regions between these extremes, e.g. around  $C_6$  and  $H_8$ , respectively, molecules range from volatile to effectively non-volatile, and thus purely C- or H-atom content is not a good predictor for volatility.

However, the fraction remaining as a function of the oxygen content of the molecule (Fig. 7c) shows the most clearly monotonically decreasing trend. The transition from non-condensing to condensing occurs at an O-atom contents of 6-7, with nitrates again being slightly more volatile than non-nitrates. Finally, Figure 7d demonstrates that the effective O:C ratio appears to be the worst predictor for volatility out of the parameters plotted. Despite O:C ratios of organic aerosol often being used as a reference for volatility, it does not correlate well with the volatility of an individual molecule. This is not surprising, as high molecular weight dimers require only a relatively small number of oxygen atoms (thus having low O:C) to condense, while small molecules with 1-3 C-atoms are unlikely to condense regardless of their O:C ratios. Therefore, the absolute oxygen number alone seems to be the best predictor for the FR.

Elemental composition can be a good indication of the possible functional groups in the molecule that ultimately determine the volatility of a compound (Kroll and Seinfeld, 2008; Pankow and Asher, 2008). This is essentially the basis of models, such as the two by Peräkylä et al. (2020) and Bianchi et al. (2019) that we tested (Fig. 6), that aim to estimate volatility directly from the easily measurable elemental composition. As such, they however always contain some intrinsic assumptions of the structure and the likely functional groups in the molecule. The Peräkylä et al. (2020) model is developed from an observed condensation of  $\alpha$ pinene products  $(C_{10}H_{16})$ , while the Bianchi et al. (2019) model is also primarily constructed with larger precursors in mind. An underlying assumption of a larger carbon frame, or a lower O:C, might explain the discrepancy between our observations and the modelled FR, as an extrapolation to molecules with less C and much higher O:C, might cause the models to fail, especially in fully capturing the observed effect the Oatom content has on the uptake (Fig. 7). Incidentally, the Bianchi et al. (2019) model predicted transition in condensation (Fig. 6) appears to be misplaced by approximately the mass difference between monoterpenes and cyclohexene. Furthermore, if the additional 4 carbon atoms that separate the two compounds are imagined into the model (replacing  $n_c$  with  $n_c+4$ ), the modified Bianchi et al. (2019) model predicts our observed cyclohexene product FR remarkably well (Fig. B2). While anecdotal, it does support our conclusion that the oxygen content of the HOM is the primary contributor to their uptake efficiency. If the Peräkylä et al. (2020) model is adjusted similarly for both the number of C-atoms (+4) as well as the number of H-atoms (+6), which is also part of the parametrisation, the result is a negligible change in the modelled FR, as the added C- and H-atom content have opposing effects on the estimated volatility.

The lesser sensitivity to oxygen content of the Peräkylä et al. (2020) model could also partly explain its comparatively slow decrease in volatility as a function of mass. Peräkylä et al. (2020) themselves already contemplated that the parametrisation might not be able to predict volatilities of products with less than 10 carbon atoms, thinking that they might depend differently on the number of oxygen atoms, for example."

**Minor Comments**

We edited to the manuscript following these suggestions:

1. Line 29. Sentence beginning with "Through oxidation..." is awkward. Consider rephrasing.

3. Line 81. Sentence beginning with "Experiments with ammonium sulfate..." is unnecessary.

5. Line 244. How were the modeled values "scaled to the range of observed remaining fractions"?
7. Line 315. This sentence is contradictory. By definition, "increased termination of RO2 radicals by NO" does not lead "to more alkoxy radicals" but rather to closed-shell alkyl nitrates. Radical propagation via reaction of RO2 with NO will produce more RO radicals, which seems to be the intent of the sentence.

For the rest of the minor comments we give our answers here:

2. Line 76. What is the detection limit of the NOx analyzer?

R12: The instrument manual states that the lower detectable limit is 0.40 ppb (60 second averaging time)

4. Figure 7. Why is there so much variability in the signal of the three species prior to ABS seed injection (0.9-1.1)? What are the uncertainties in the measured signals of the detected species and their relative changes as a function of NOx mixing ratio and ABS seed concentration?

R13: The variation of 10% in this plot is mainly due to counting statistics. As can be seen from the new Fig. 1, the signals of the plotted compounds are on the order of a few counts per second, and this variability is to be expected. For the relative changes with NOx and seed, we took longer averages, and therefore the variability due to counting statistics will be much smaller. The largest source of uncertainty in general in our data is likely from wall interactions of SVOC-type compounds. We now explicitly added the sentence:

"This type of behaviour is likely the largest source of uncertainty in our data, and the potential error will be different for all molecules and therefore difficult to quantify in any detail. A rough estimate is that this would at most cause an error of 10-20 % to the relative changes reported above for the relatively fast seed addition experiments."

6. Line 250. A discussion of the suitability of using an activity coefficient of unity and the potential impacts of this assumption should be included.

R14: We have now extended this part to read: "We assumed an activity coefficient ( $\gamma$ ) of 1, so that  $C^*=\gamma C^0=C^0$ . Given the small precursor molecules in our study, the formed SOA is expected to have very high O:C ratios, just like the majority of potentially condensing molecules we measure in the gas phase. Under these circumstances, and considering the low RH of our experiments, an activity coefficient close to unity is likely to be a valid assumption (Donahue et al., 2011)."

**Referee #2**

Raty et al. generated highly oxygenated organic molecules (HOM) from the ozonolysis of cyclohexene in an environmental chamber. They separately characterized the effects of adding NO and ammonium bisulfate (ABS) seed particles on cyclohexene HOM composition. HOM were detected with a time-of-flight chemical ionization mass spectrometer using nitrate reagent ion. Following NO addition to the chamber, the relative abundance of C6H8O9, C6H9NO9, C12H19NO11 (and other HOM) increased, especially nitrogen-containing HOM. The abundance of C6H8O7, C12H20O9 and other HOM decreased. Following ABS addition to the chamber, signals at C6H10O4, C6H10O6, and C6H10O8 – and other low-volatility HOM - decreased due to the increased condensation sink. A model was used to relate the fraction of HOM remaining in the gas phase to its effective saturation concentration (C\*). Overall, the experiments are well motivated from the perspective of trying to better understand (1) the composition of molecules that contribute to new particle formation and (2) the effects of NOx and condensation sink perturbations on NPF. However, in my opinion, the novelty and atmospheric significance of the results are uncertain the way they are currently presented. The comments below should be implemented into a revised manuscript before I would support eventual publication in ACP.

R15: Before addressing the specific comments by the referee, we note that both referees request more chemical insights and conclusions from this work. As stated also in R1, our aim is more to provide a comprehensive set of data and analysis from the studied cyclohexene system, without going into details on specific molecules or mechanisms. In order to make the aims and scope of our manuscript more clear, and in such a way avoid misunderstandings or misconceptions for the reader concerning what the manuscript will contain, we have made a few amendments:

- 1. We have classified this manuscript as a "Measurement report", as this article type better corresponds to our aims.
- 2. We have rewritten the last paragraph of the introduction and the 3rd sentence of the abstract to better outline our aims and approach. Also the conclusions section has been considerably updated.

Many sections of the text were also edited for a hopefully added clarity.

We hope that in this way we can avoid giving readers wrong expectations on the content of our results.

**General Comments**

1. Peräkylä et al. (2020) describe a similar set of experiments with a different precursor (a-pinene). In that study, in general, I felt that the analysis was clearer and more thorough than was presented here. At the very least, because a study of cyclohexene-derived HOM is motivated here as "a surrogate for monoterpenes with an endocyclic double bond" to assess "how applicable earlier volatility parameterisations are on the cyclohexene system" a revised manuscript should incorporate parallel analyses to what were presented in the companion Peräkylä et al. (2020) manuscript. For example, Figure 8 in Peräkylä et al. (2020) shows the calculated C\* values of C10H16Ox HOM that were calculated from seed perturbation experiments. Actual C\* values of cyclohexene-derived HOM are never plotted or discussed here as far as I can tell. Also, Figure 7 of Peräkylä et al. (2020) shows a scatter plot comparing the modeled versus measured fraction

remaining (FR) of a-pinene-derived HOM. In my opinion this is a much clearer presentation than Figure 8 in this manuscript.

R16: Our aim was not initially to duplicate the Peräkylä et al. (2020) analyses, but practical limitations also restricted us from doing all the similar analysis steps. In particular, the data quality and quantity was much higher in the study by Peräkylä et al. (2020). We used a lower resolution instrument (HTOF with mass resolving power ~5000 Th/Th, compared to LTOF ~10 000 Th/Th in Peräkylä et al. (>13 000 Th/Th specifically in their mass range)), meaning that we are able to reliably separate fewer ions from the spectra (therefore the need for the FitFactor approach).

In addition, cyclohexene produces less HOM peaks to begin with than the larger and structurally more complex a-pinene. Finally, we also had fewer successful repetitions of different parameter changes. For these reasons, we took a different approach to this study, trying to maximize the amount of information that we could extract from the data, but at the same time avoiding to draw too far-going conclusions, or providing new parametrizations. As the reviewer also notes, we did not try to estimate C\* values for our HOM. Partly for reasons discussed above, and also because estimates from the FR require assuming that the uptake is driven purely by vapour pressure, and not e.g. by reactive uptake.

We added the following sentence to the text, as the information on the specific TOF used was initially missing:

"[...Time-Of-Flight (ToF) mass spectrometer,] which in our setup was an HTOF that has a mass resolving power of ~5000 Th/Th."

Concerning the final comment regarding Figure 8:

We wanted to include a fraction remaining vs. mass figure in our analysis, and since we had a limited amount of data points to show, it gave a good opportunity to have the modelled data points in this figure as well. As the number of these "good fit" data points is not very large, we think the main messages are easy to observe from this figure. For example, that the models overpredict the volatility. We had also plotted the suggested types of observation vs. model figures as were shown by Peräkylä et al. (2020). However, as the models worked so poorly, we initially opted not to include them. The plots are provided below (a:Peräkylä et al., b:Bianchi et al.). We have now included these plots in the appendix, as they indeed very clearly show the extent of the discrepancy.

2. After setting up Section 2.3 for a discussion of the cyclohexene ozonolysis mechanism, it transitions to a more abstract/general discussion after L136. I think it would be better to focus the discussion on what happens to the C6H9O4 peroxy radical, such as the specific autooxidation and/or RO2-RO2 reactions that lead to some of the major HOM products, i.e. C6H8O7, C6H8O9, and C12H20O9, which have already been identified in earlier studies (e.g. Rissanen et al.). A figure with a reaction scheme showing these autooxidation steps would also be useful. Reframing the discussion around specific autooxidation steps that lead from C6H9O4 to HOM, along with a reaction scheme, allows for a more direct transition to the results and discussion of the NOx and condensation sink perturbation studies.

R17: The beginning of this section was indeed fairly inconsistent and has now been edited. In general, we refer to our response R1. As this paper does not target mechanistic understanding (and there already are several that do), we found that a more suitable approach was to remove the initial set up with the mechanistic detail, and to keep the introduction more general. As noted also in R15, too much mechanistic detail here might mislead the reader into expecting mechanistic results, which are not presented in this paper.

3. The way the paper is currently written, the relative roles of RO2 + NO and RO2 + NO3 reactions in generating the results that are presented/discussed in Figure 3 and Section 3.1 are not clear: [NO] is below detection limit, and [NO3] is not constrained by measurements and/or modeling. At the least, a photochemical box model simulation (e.g. KinSim or similar, see Peng and Jimenez, 2019) with the relevant COALA chamber conditions, reactions and rate coefficients would be appropriate here, perhaps as an appendix. Because the use of cyclohexene is motivated as a simple surrogate for monoterpenes, it would also be appropriate to add another reaction scheme to Section 3.1 that explains the increases or decreases in HOM observed in Figures 3-4 following perturbation by NO (and/or NO3).

R18: Concerning the kinetic modelling, we refer to our response R4. The role of RO2 + NO would require knowledge of the rate constants for both RO2 + NO and RO2 + RO2, which will have uncertainties of orders of magnitude. As such, we prefer to refrain, as stated in R15, from explicitly suggesting branching ratios or relative weights for different reaction pathways.

**Minor/Technical Comments**

4. L67: The authors state: "The resulting steady state concentration of ozone was approximately 18 ppb, while a rough estimate for cyclohexene concentration was about 100 ppb [...] estimated from the difference in ozone concentration with and without cyclohexene". What was the ozone concentration prior to cyclohexene addition? This would be useful for any readers that might try to reproduce the experimental conditions described here. To what extent is the HOM composition, e.g. the monomer:dimer ratio, influenced by [cyclohexene]:[ozone]?

R19: We added now to the text that the ozone concentration was roughly 25 ppb before adding cyclohexene. We had also initially rounded our cyclohexene estimate to 100 ppb, but amended it now to 70 ppb.

"The ozone concentration without cyclohexene in the chamber, was on average 25 ppb (23-27 ppb), and when cyclohexene was added the steady-state ozone concentration was approximately 18 ppb (16-19 ppb).

The cyclohexene concentration was not measured and therefore had to be estimated from the drop in ozone concentration at its introduction. This produced a rough estimation of 70 ppb of cyclohexene."

Regarding the monomer:dimer ratio, we do not have enough variability in our conditions to analyse this. For more studied systems, like a-pinene + ozone, the HOM formation has seemed to be dependent only on the oxidation rate, i.e. [VOC]\*[O3], not the ratio of the two (Ehn et al., 2014).

5. L77: Please explain why 9 ug/m₃ loading of ABS, with corresponding condensation sink of ~0.085 s-₁, was chosen for the seed perturbation studies. Additionally, ozonolysis of ~100 ppb cyclohexene presumably generates some SOA given reported SOA mass yields of approximately 0.15 – 0.20 (Keywood et al., 2004). If that's the case here, what is the condensation sink of homogenously nucleated cyclohexene ozonolysis SOA relative to the added ABS seeds?

R20: Controlled seed generation of size-selected particles into the steady-state chamber is challenging, and in reality a larger seed concentration was planned. However, the resulting 9 ug/m3 loading of ABS was assumed to be sufficient, when the experiments were conducted. A higher aerosol loading would have given clearer results and thus been beneficial for data interpretation.

As we used a steady state chamber, the reacted VOC concentration was not 100 ppb (we also adjusted our estimation for cyclohexene to 70 ppb), as the referee suggests. The reported ozone drop of about 7 ppb means that during the 50 min average residence time in the chamber, around 7 ppb of cyclohexene reacted with ozone. Another max. 7 ppb may have reacted with OH, generated from the ozonolysis. A rough estimate would thus be that we are reacting roughly 14 ppb cyclohexene, which is equivalent to roughly 50  $\mu$ g/m3, during an average residence time in the chamber. The yields quoted by the referee are for much higher loadings (e.g. Keywood et al), and under our conditions a more likely value is well below 0.05 (i.e. maximum expected SOA of 2.5  $\mu$ g/m3). The measured SOA mass was just less than 1  $\mu$ g/m3, and given that some fraction of the HOM will have condensed on the walls, our SOA observations compare well with earlier studies.

The above calculations were done for the case with seed aerosol. Without seed, the majority of HOM will condense onto chamber walls, and the SOA mass will be negligible.

6. Figure 1 and some of the accompanying discussion could probably be moved to an appendix or supplement.

**R21: This part was moved to an appendix, as suggested.**

7. Figure 2 and 3 could be combined into a single 2-panel figure to facilitate easier comparison. I would also consider simply adding a 3rd panel showing the mass spectrum of cyclohexene HOM following the addition of 0.085 s-1 ABS condensation sink, and removing Figure 7. Unlike Figure 4, which shows the change in HOM across a continuum of NOx values, there is only one ABS condensation sink value, so there are no meaningful trends to show in Figure 7 that couldn't be more simply presented as a mass spectrum to directly compare with Figure 2.

R22: The two figures were combined, as suggested. However, the suggested 3rd panel is not very informative, as shown below. The blue spectrum shown below is the +NOx spectrum from the manuscript, while the red spectrum shows the signals when ABS is also injected into the chamber. The spectra are dominated by non-condensing species, and therefore it is hard to see the changes in the HOM region.

We agree about figure 7 that it does not in itself provide very much new information. The main reason to include it is in order to make clear how the FR is calculated, so that a reader can more easily follow the analysis. For this reason, we would prefer to keep this figure in the manuscript.